# Strategy to mitigate substrate inhibition in wastewater treatment systems

Beiying Li[1], Conghe Liu[1], Jingjing Bai[1], Yikun Huang[1], Run Su[1], Yan Wei[2] & Bin Ma ®[1] ✉

Global urbanization requires more stable and sustainable wastewater treatment to reduce the burden on the water environment. To address the problem of substrate inhibition of microorganisms during wastewater treatment, which leads to unstable wastewater discharge, this study proposes an approach to enhance the tolerance of bacterial community by artificially setting up a non-lethal high substrate environment. And the feasibility of this approach was explored by taking the inhibition of anammox process by nitrite as an example. It was shown that the non-lethal high substrate environment could enhance the nitrite tolerance of anammox bacterial community, as the specific anammox activity increasing up to 24.71 times at high nitrite concentrations. Moreover, the system composed of anammox bacterial community with high nitrite tolerance also showed greater resistance (two-fold) in response to nitrite shock. The antifragility of the system was enhanced without affecting the operation of the main reactor, and the non-lethal high nitrite environment changed the dominant anammox genera to *Candidatus* Jettenia. This approach to enhance tolerance of bacterial community in a non-lethal high substrate environment not only allows the anammox system to operate stably, but also promises to be a potential strategy for achieving stable biological wastewater treatment processes to comply with standards.

By 2050, over two thirds of the global population will live in cities[1,2]. Urbanization has led to the pollution of water environments with a variety of pollutants from human sources, including nutrients and pathogens[3]. Eutrophication, caused by nutrients such as nitrogen (N) and phosphorus (P), is widespread in many regions worldwide, particularly in developing countries[4]. Rapid global urbanization is placing an unprecedented burden on the water environment[5–7]. More effective wastewater treatment strategies are required to address rising pollutant levels while minimizing energy emissions to meet peak carbon and carbon-neutral targets[8,9].

Microbiological processes are essential to wastewater treatment strategies[10] as they remove pollutant elements such as carbon, nitrogen, phosphorus and sulfur[9,11–14]. Taking biological nitrogen removal as

an example, the nitrogen removal pathways in wastewater treatment include nitrification and denitrification reactions[15]. The process of nitrification involves the oxidation of ammonia to nitrate by ammonia-oxidizing bacteria, ammonia oxidizing archaea and nitrite-oxidizing bacteria[16,17]. Denitrification, on the other hand, completes the nitrogen cycle by converting nitrate into nitrogen gas[18,19]. Anaerobic ammonia oxidation (Anammox) is an emerging microbial processes to reduce energy consumption and enhance the efficiency of nitrogen removal[20,21].

Substrates are necessary for microbial growth in wastewater treatment processes. However, high substrate inhibition is usually triggered when the concentration exceeds a certain threshold[22,23]. For instance, high concentrations of ammonia, nitrite and hydrogen

[1]Key Laboratory of Agro-Forestry Environmental Processes and Ecological Regulation of Hainan Province, School of Environmental Science and Engineering, Hainan University, Haikou 570228, China. [2]State Key Laboratory of Marine Resources Utilization in the South China Sea, Hainan University, Haikou 570228, China. ✉ e-mail: mabin@hainanu.edu.cn

sulphide can be toxic to wastewater treatment microorganisms[24,25]. Increased substrate concentrations can reduce microbial activity, hindering the conversion of the substrate. The decreased conversion can cause substrate accumulation around the microorganisms, resulting in stronger toxic effects[26]. Substrate inhibition can disturb the balance of the system through negative feedback, thereby reducing the efficiency of the wastewater treatment process[27,28].

The existing strategies for dealing with substrate inhibition revolve around avoiding or decreasing inhibition and recovery after inhibition in the wastewater, involve reducing the concentration of substances and adjusting operating conditions[29–32]. The main principle of these measures is to decrease the contact between the microbial community and the high substrate, making the community more vulnerable to inhibitory substrates[33]. However, substrate fluctuations are a global situation in wastewater treatment and vary over time due to factors such as industrial discharges, seasonal variations and population fluctuations[34]. The field survey revealed that high-substrate is unavoidable[35,36].

When substrate fluctuations cannot be controlled, another potential strategy involves enhancing resistance of microbial community. The increased resistance has the potential to maintain or even increase the system treatment efficiency and stability in the presence of substrate stress. This mechanism is analogous to antifragility[37]. To achieve the antifragility wastewater treatment system by enhance tolerance of microbial community, a feasible hypothesis was proposed to create a non-lethal high substrate inhibition environment.

In the wastewater treatment, anammox is the most promising low-energy wastewater treatment technology[38,39]. It convert nitrite and ammonia simultaneously into nitrogen under anaerobic conditions without organic carbon source[40,41], which can save up to 60% of energy consumption for aeration and eliminate the requirement for external organic carbon sources[42]. The promotion of anammox is consistent with the objectives of peak carbon and carbon neutrality[43,44]. Anammox was a process driven by autotrophic bacteria, including four genera (*Kuenenia*, *Brocadia*, *Anammoxoglobus* and *Jettenia*) commonly found in activated sludge[45]. The enrichment and physiology work demonstrated that nitrite is an important substrate for anammox bacteria, but nitrite with a high level can also act as an inhibitor[46,47]. Despite taking measures such as reducing nitrite concentration and adjusting pH to minimize nitrite inhibition, half of the wastewater treatment plants (WWTPs) still exhibited high levels of nitrite accumulation during the field survey[48].

In this study, nitrite inhibition of anammox bacterial community was used as an example, developing a sidestream nitrite treatment unit located outside the anammox systems (Fig. 1). A portion of the anammox sludge was transported from the system to this sidestream unit with a high nitrite concentration, and then returned to the anammox system. To verify the feasibility of improving the nitrite tolerance of anammox bacterial community in a non-lethal high nitrite environment, firstly, the response to different nitrite levels of anammox bacterial community was investigated to evaluate the variation in nitrite resistance. Secondly, the operational stability of the anammox system during substrate fluctuations was examined by analyzing the nitrogen removal performance under nitrite shock. Furthermore, the nitrogen conversion rate of the anammox system was measured to evaluate the effect of sidestream nitrite treatment on the anammox system. Finally, the effect of a high nitrite environment on the community was explored through 16S rRNA and metagenomics sequencing. This study contributes to the understanding of the resistance of anammox bacterial community to nitrite and further proposed a potential strategy to cope with substrate inhibition in wastewater treatment engineering, promoting the stability of wastewater treatment systems under substrate fluctuations.

## Results

### Improved the tolerance of anammox bacterial community through high-nitrite exposure

To explore the nitrite tolerance of anammox bacterial community, the SAA was detected in different nitrite levels after exposure to high-nitrite environment. The SAA of anammox bacterial community from UASB1 and UASB2 were expressed by SAA1 and SAA2, respectively. Trends in SAA1 and SAA2 at different nitrite concentrations were shown in Fig. 2, and the MLSS were 7.28 g·L$^{-1}$ and 5.35 g·L$^{-1}$, respectively.

The results showed that exposure to high nitrite concentrations, but not lethal, enhanced the tolerance of anammox bacterial community. The SAA1 reached its peak when the nitrite-to-ammonia ratio was 1.0 (with a nitrite concentration of 10 mg·L$^{-1}$). When the ratio increased to 3.0 (with a nitrite concentration of 30 mg·L$^{-1}$), SAA1 decreases by over 69%. And SAA1 decreases by 80.81% when the ratio reaches 4.0.

Comparing the results, SAA2 was much higher than SAA1 throughout all five nitrite concentrations. Anammox bacterial community that had been previously exposed to high-nitrite were able to maintain their activity while the nitrite-to-ammonia ratio increased to 3.0. Furthermore, even when the nitrite concentration was up to 40 mg·L$^{-1}$, SAA2 was 24.71 times higher than that of SAA1, and the nitrite removal rate of anammox bacterial community in UASB2 was 20.15 times higher than that of UASB1.

These results indicate that anammox bacterial community in UASB2 were more tolerant to high nitrite levels. Anammox bacterial

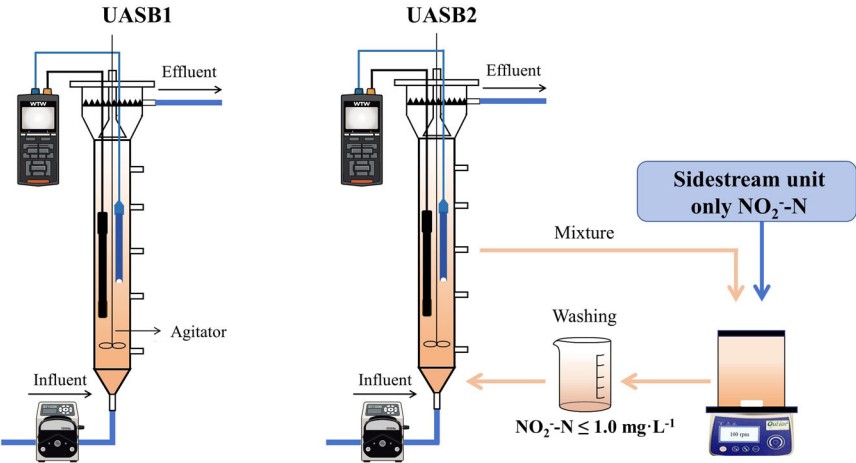

**Fig. 1 | Device details of two UASB reactors.** UASB1 was the control reactor, UASB2 was equipped with a sidestream unit separately.

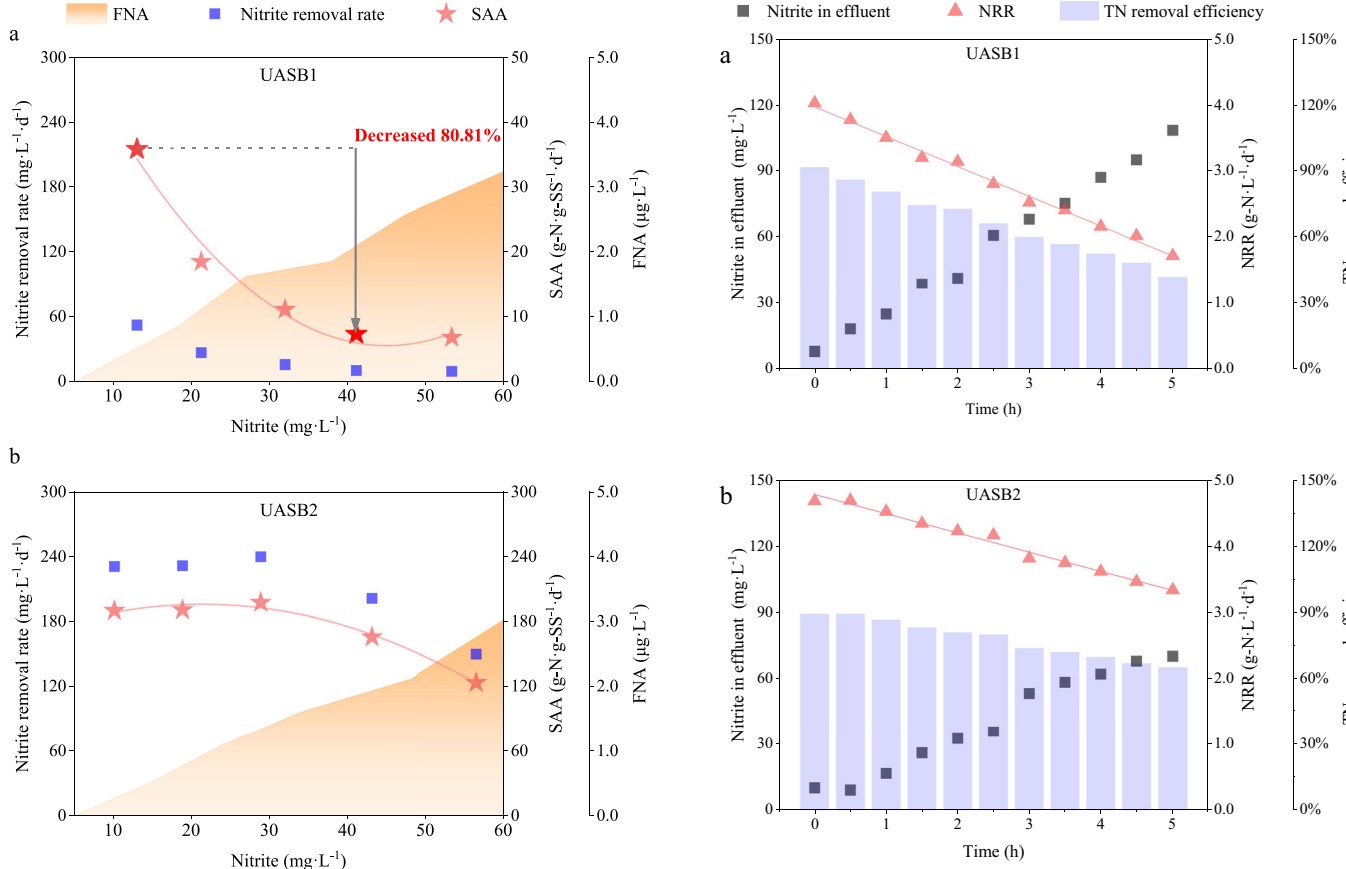

**Fig. 2 | The nitrite tolerance of anammox bacterial community was explored by detecting the nitrite removal rate and SAA in different nitrite levels.** (**a**) was UASB1 reactor and (**b**) was UASB2 reactor.

**Fig. 3 | The nitrogen removal performance of effluent nitrite concentrations, NRR and TN removal efficiency under nitrite shock.** (**a**) was UASB1 reactor and (**b**) was UASB2 reactor.

community had higher activity after exposure to nitrite, even when the nitrite concentration exceeded the theoretical anammox stoichiometric ratio. The tolerance of microbial community to inhibitory substrates was enhanced by a non-lethal inhibitory environment.

### Increased the shock resistance of anammox systems through high-nitrite exposure

In order to investigate the resistance of the anammox system to high nitrite concentrations, a sudden increase of influent nitrite concentration from 80 mg·L⁻¹ to 200 mg·L⁻¹ to achieve the nitrite shock. The nitrogen removal performances, such as effluent nitrite concentration, NRR and TN removal efficiency was shown in Fig. 3.

Anammox systems with the sidestream nitrite treatment unit exhibited greater operational stability during the nitrite shock test. Within 5 hours, NRR1 was decreased by 57.35%, while NRR2 was only decreased by 28.85%. The trend of TN removal efficiency was consistent with the NRR. In UASB1, the decrease in TN removal efficiency was more than double that of UASB2. The slower decline in nitrogen removal performance indicated that the anammox system was more antifragile to high nitrite concentrations, suggesting that UASB2 exhibiting twice the level of antifragility compared to UASB1.

Nitrite concentrations in UASB1 was increased from 7.92 mg·L⁻¹ to 108.51 mg·L⁻¹ over time was caused by the lower nitrite tolerance of anammox bacterial community. As the nitrite concentration increased within the reactor, it diminished the activity of anammox bacterial community and the ability to remove nitrite, consequently leading to further nitrite accumulation and stronger nitrite inhibition. This negative feedback mechanism caused NRR1 to decline significantly faster than NRR2. The increase rate of nitrite concentration in UASB2

was significantly slower. This can be attributed to the anammox bacterial community in UASB2 being able to maintain greater activity at high nitrite concentrations. It enabled the simultaneous removal of ammonia and nitrite, providing the anammox system a more stable nitrogen removal performance. In conclusion, exposing the microorganisms to high substrate concentrations in the sidestream unit can improve the stability of the system to substrate fluctuations.

### Enhanced antifragility of anammox system through high-nitrite exposure

In order to evaluate the effect of high nitrite treatment on main system, the nitrogen removal performance of UASB2 reactor was investigated during the operation of the sidestream unit. The analyses showed that there was an effect of the high nitrite treatment on the anammox system, as evidenced by fluctuations in the nitrogen removal performance. However, this effect did not reappear during the second high nitrite treatment. The nitrogen removal performance of the UASB1 reactor was shown in Supplementary Fig. 1.

A portion of the anammox sludge was transported from the system to the sidestream unit for exposure to high nitrite on day 34 (phase II). After one day of operation, the effluent nitrite concentration of UASB2 immediately increased from 5.44 mg·L⁻¹ to 30.47 mg·L⁻¹ (Fig. 4). During ten days of operating the sidestream nitrite treatment unit, the average effluent nitrite concentration in UASB2 was 28.03 mg·L⁻¹. Consequently, there was a significant decrease in the NRR of UASB2, amounting to approximately 56.04%. These results emphasize that the first operation of the sidestream nitrite treatment unit had an impact on the nitrogen removal performance of the reactor.

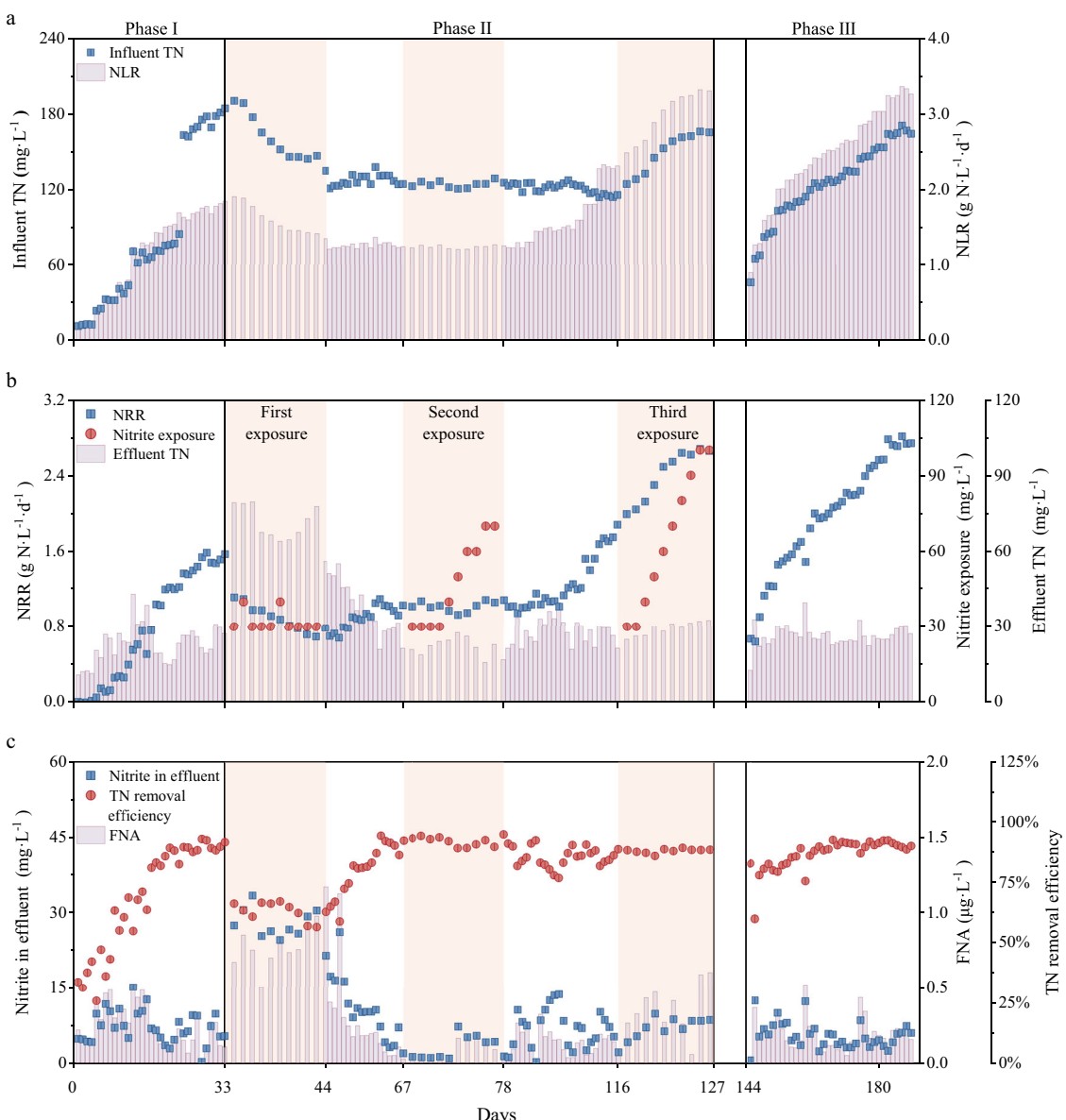

**Fig. 4 | Reactor performances of the UASB2 reactor. a** Influent TN concentrations and NLR. **b** Effluent TN concentrations, nitrite exposure concentrations and NRR. **c** Effluent nitrite concentrations, FNA and TN removal efficiency.

The second nitrite exposure operation in the sidestream unit was started on day 68. The nitrite effluent concentration of UASB2 remained below 10 mg·L$^{-1}$ throughout the second operation, and the NRR of UASB2 was consistent that before nitrite exposure. Compared to the first operation, the nitrite exposure from the sidestream unit had a significantly lower impact on the UASB2 reactor.

The sidestream nitrite treatment unit then began its third operation on day 117. Remarkably, during this operation, the TN removal efficiency was maintained at 88.5% despite the nitrite exposing concentration in the sidestream unit gradually increased from 30 mg·L$^{-1}$ to 100 mg·L$^{-1}$. Anammox bacterial community were exposed to a higher nitrite concentration during the third operation, the NRR of UASB2 increased significantly by 33.3%. The anammox system are not only able to withstand nitrite stress, but also exhibit antifragility by displaying high nitrogen removal efficiency in the presence of stress.

The exposure of the anammox bacterial community to high nitrite concentrations in the sidestream unit played a crucial role in enhancing their nitrite tolerance. Upon returning to the normally operating UASB2 reactor, the anammox bacterial community had the

opportunity to recover under low nitrite concentrations and simultaneous presence of ammonia. This non-lethal exposure and timely recovery of anammox bacterial community likely acquired a heightened resistance and improved nitrite tolerance, which potentially enhanced the ability to withstand and adapt to varying nitrite levels.

### High-nitrite environments alter population diversity

The Ace and Chao indices were used to measure species richness. Lower values in these estimators correspond to decreased species richness. Concurrently, the Simpson and Shannon indices were employed to assess species evenness, where a lower Simpson index and a higher Shannon index indicate higher species evenness[49,50]. The microbial diversity in seed sludge and UASB1 was higher than UASB2. Higher diversity means that there are more complementary units adapted to nitrite stress, which could work together to cope with stresses[51]. In this way, the anammox bacterial community were able to adjust and repair more quickly when exposed to high nitrite concentrations, enhancing the adaptation of the community. There was a slight decrease in microbial richness and evenness in UASB2, and this

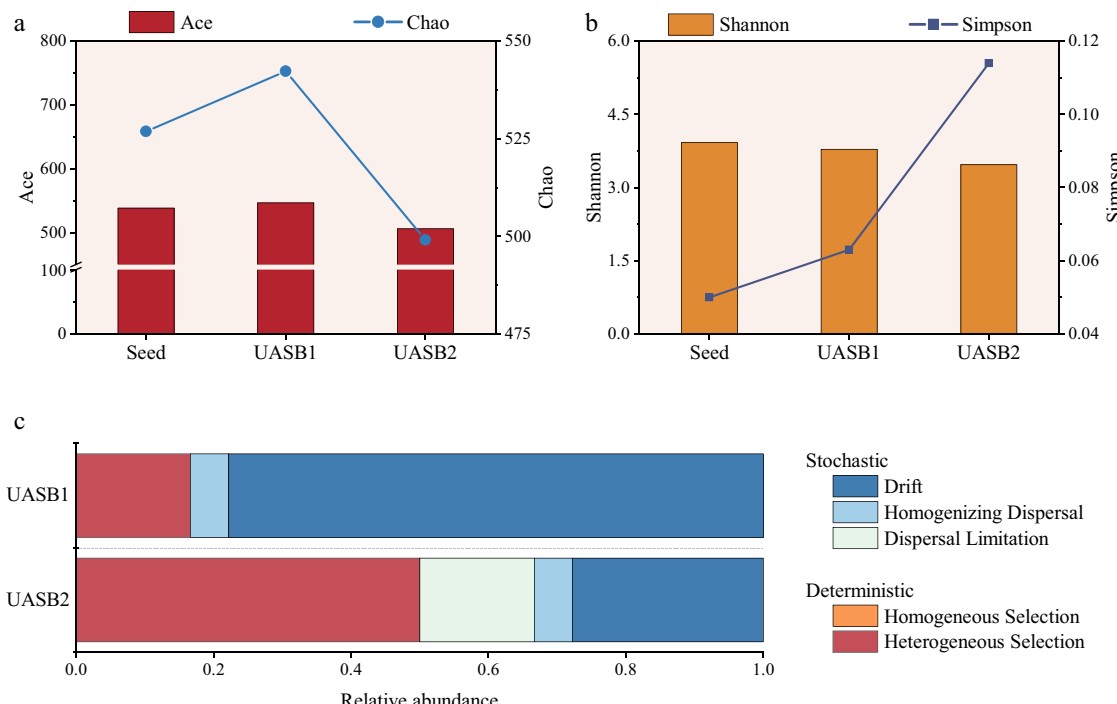

**Fig. 5 | Characteristics of population diversity. a** Species richness expressed via Ace and Chao estimator. **b** Species evenness expressed via the Shannon and Simpson indices. **c** The summary of assembly processes.

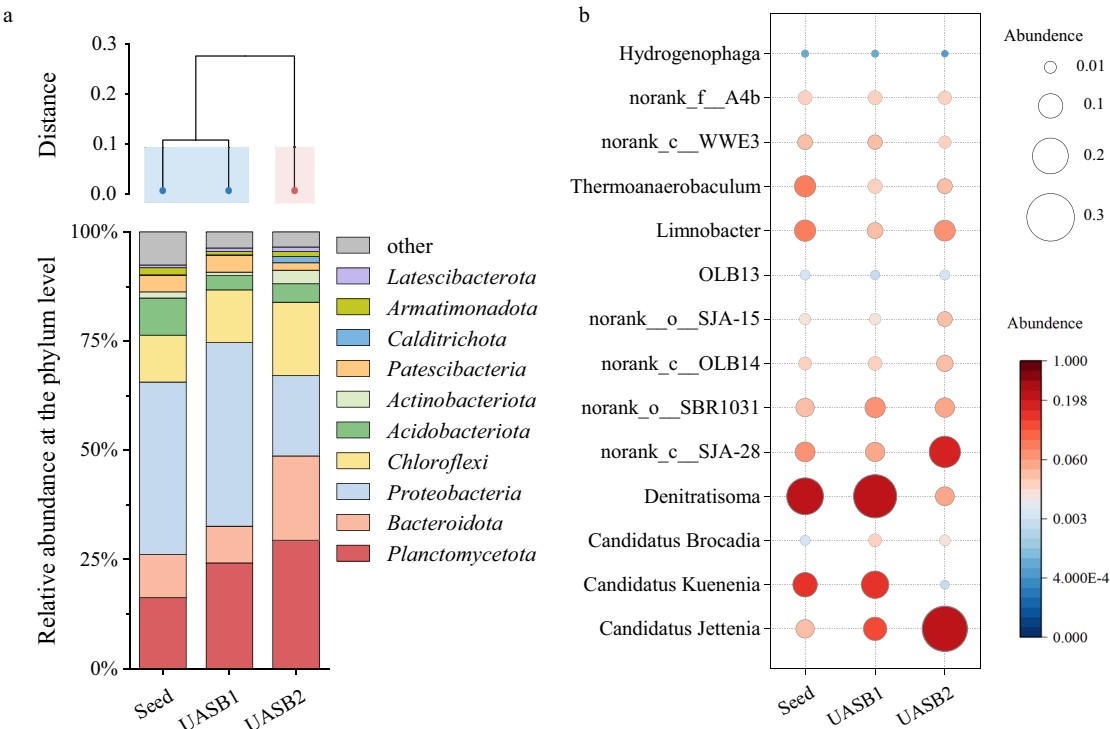

**Fig. 6 | The structure of microbial communities. a** Hierarchical clustering tree and distribution of the bacterial community on phylum level. **b** Analysis of bacterial community abundance at the genus level.

decrease was a response of the microbial community to high nitrite stress (Fig. 5).

The variation in the community assembly process was further investigated by comparing the βNTI index, and the results of the βNTI analyses were used to calculate the proportions of deterministic and stochastic processes (Fig. 5c). The results showed that heterogeneous

selection dominated for microbial communities in UASB2, with basic abiotic parameters as the driving factors[52]. Nitrite concentration was probably the main driver in this study. Hierarchical clustering analysis were used to compare bacterial communities from different samples, as shown in Fig. 6a. The analysis revealed that the UASB2 sludge samples were dissimilar from the other samples based on phylogeny.

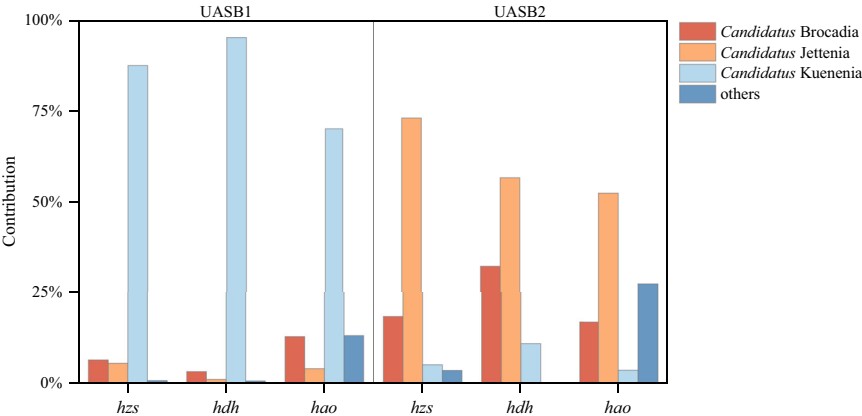

**Fig. 7 | The contribution of anammox bacterial community to key functional genes.** Left panel: Control group without nitrite addition (UASB1). Right panel: Treatment group with nitrite addition (UASB2).

The observed dissimilarity in UASB2 may be attributed to microbial responding environmental stressors, such as nitrite exposure, prompted a deeper investigation into the microbial community structure.

### High-nitrite environments shift the structure of microbial communities

The study found that the abundance of three phyla, *Planctomycetes*, *Chloroflexi* and *Bacteroidetes*, increased in UASB2 as compared to UASB1, while there was a significant decrease in the abundance of *Proteobacteria* (Fig. 6a). *Planctomycetes* phylum contained three types anammox genera, *Candidatus* Kuenenia, *Candidatus* Brocadia and *Candidatus* Jettenia (Fig. 6b). Among these, *Candidatus* Kuenenia was the dominant genera in the seed sludge and UASB1 sludge, the relative abundance was 9.93% and 12.62%, respectively. The dominance of *Candidatus* Kuenenia can be attributed to its k-strategist characteristics, making it more competitive under conditions of low nitrogen concentrations[53–55].

After nitrite exposure, the main anammox bacterial genera were *Candidatus* Jettenia, accounting for 28.26% in UASB2, while the remaining two genera accounted for less than 0.5%. *Candidatus* Kuenenia was decreased abruptly due to their susceptibility to fluctuating substrate conditions[34]. This transformation in response to nitrite shock potentially contributed to the antifragility of UASB2 reactor. Since *Candidatus* Jettenia may possess superior adaptive advantages in high nitrite concentrations compared to other anammox genera, it was able to adapt well to influent fluctuation. The variation of anammox bacterial community in UASB2 was consistent with the diversity. The adapted groups grew and multiplied, and the non-adapted groups withered and died, changing the proportions of different genera in the anammox bacterial community.

*Denitratisoma* and *norank_c_SJA-28* are the two genera in the UASB2 system with relatively significant abundance variations, except anammox genera. Both *Denitratisoma* and *norank_c_SJA-28* have the ability to participate in denitrification reactions[56–58], which was important in anammox systems[59,60]. Nevertheless, *Denitratisoma* decreased in abundance at high nitrite concentrations[61]. In some cases, organisms in the *SJA-28* lineage encoded an ammonia-forming cytochrome c nitrite reductase of the *nrf*AH-type involved in nitrite reduction[62]. That may allow *SJA-28* survive higher nitrite concentrations, which may explain the increased abundance in UASB2.

More analyses of the different bacterial communities in UASB1 and UASB2 in the Supplementary Information. Differences of bacterial communities in UASB2 may be due to different responses to nitrite-induced stress. Further exploration of microbial functions could provide insights into changes in community distribution.

### High-nitrite environment transformed the community contribution to functional genes

The anammox process involves the conversion of nitrite and ammonia to $N_2$ gas through three possible pathways. This conversion process relies on the synergistic action of three key functional enzymes: hydrazine synthase (HZS), hydrazine dehydrogenase (HDH), and hydroxylamine oxidoreductase (HAO)[63]. HZS catalyses the conversion of NO from nitrite to hydrazine ($N_2H_4$) with $NH_4^+$. The $N_2H_4$ generated is then oxidized by HDH and eventually converted to $N_2$[64]. HAO was believed to reconvert any hydroxylamine ($NH_2OH$) that escaped from the HZS complex back into NO, thereby detoxifying this inhibitor $NH_2OH$ and ensuring the normal progression of the anammox process[65,66].

The genes producing these enzymes were detected by metagenomic analysis to investigate the contribution of anammox bacterial community to key functional genes. It was found that the functional genes *hzs*, *hdh* and *hao*, encoded by *Candidatus* Kuenenia in the UASB1, were highly abundant, contributing 91.76%, 92.37%, and 80.82%, respectively (Fig. 7). After the nitrite treatment, the abundance of functional genes encoded by *Candidatus* Jettenia increased significantly and became the largest contributing anammox genus. It indicated that the anammox process was dominated by *Candidatus* Jettenia bacteria in UASB2, which was consistent with the change in community structure.

Variations in other genes related to nitrogen metabolism were shown in Supplementary Fig. 4. and Supplementary Fig. 5.

### Discussion

The non-lethal high substrate environment could enhance the nitrite tolerance of anammox bacterial community, as demonstrated in the following two aspects: (1) Anammox bacterial community exhibited 24.71 times greater tolerance to nitrite, as shown by the higher SAA at nitrite concentration was 40 mg·L$^{-1}$; (2) Anammox systems displayed greater resistance during sudden nitrite shock from 80 mg·L$^{-1}$ to 200 mg·L$^{-1}$.

The enhancing antifragility of the anammox system was attributed to the non-lethal exposure of nitrite in the sidestream unit, this operation resulted in shifted the dominant genera of the anammox bacterial community. Microbial communities have developed a variety of strategies to survive in toxic environments, and the shift of nitrite tolerance within the anammox dominant genus was a protective mechanism. *Candidatus* Jettenia has been identified as the dominant anammox population in response to increased substrate load, as a high nitrogen concentration may be more favorable for *Candidatus* Jettenia[67,68]. A study has found that *Candidatus* Jettenia was detected as the influent substrate concentration increased, and its abundance

increased by approximately 0.51% within 13 days[69]. In this experiment, through nitrite exposing in the sidestream unit repeatedly and at a maximum nitrite concentration up to 100 mg·L$^{-1}$, the proportion of *Candidatus* Jettenia increased and became the dominant anammox genera. *Candidatus* Jettenia might possess a higher tolerance to nitrite. High nitrite concentrations in the sidestream unit inhibited the growth of other anammox genera such as *Candidatus* Brocadia. Meanwhile, *Candidatus* Jettenia was in a better position to survive than other anammox genera, increasing abundance.

During practical wastewater treatment, it has been observed that the nitrogen removal efficiency of the anammox system was still reduced following repetitive occurrences of high nitrite concentrations[27,28]. This may be because reducing nitrite concentration immediately was the primary operation when high nitrite concentration appeared[70,71]. This operation is not sufficient to select for another anammox genus into one that is more adapted to high nitrite concentrations, and also making the anammox community more vulnerable towards fluctuations in nitrite concentrations.

Combined with the results of this study, a sidestream unit for nitrite treatment could be added in the biological nitrogen removal via anammox in wastewater treatment system. A portion of the anammox sludge were channel into this unit, and then returned to system after nitrite treatment, contributing to increase the antifragility of the anammox system to nitrite. The nitrite used in the sidestream unit can be produced from anaerobic digester liquor of wastewater treatment plants[72,73], and nitrite as a green and renewable chemical can be treated in the subsequent anammox reactor before its final discharge[74,75]. However, the addition of a backflow device for nitrite treatment will increase operating costs. Additions can be made based on the process and operational goals of the actual WWTPs. Depending on local conditions, they may vary from region to region and country to country.

Creating non-lethal high substrate inhibition environment could be used to improve resistance not only to nitrite, but also has the potential to be used for other substrates, such as ammonia and sulfur are present in wastewater treatment processes[76,77]. Variations in microbial tolerance to different substrate inhibition in wastewater treatment have been investigated extensively. For instance, *Nitrosomonas* cluster 7 has a stronger tolerance to high ammonia than other ammonia oxidizing bacteria[78], sulphur-oxidising bacteria *Chromatiaceae* are more tolerant of sulfur (thiosulphate) than *Thiobacillus*[79], *Ca.* Accumulibacter clades IID involved in denitrifying phosphorus removal via nitrite pathway showed strong tolerance to nitrite than clades IIC[80]. Consequently, the non-lethal high substrate could use to increase tolerance of functional microbial communities. The creation of a non-lethal high substrate environment in this study was achieved using the sidestream treatment unit. This may be a feasible strategy to enhance the antifragility of the system by improving the tolerance of the microbial community without affecting normal operation. It is a potential approach to ensure that the performance of the wastewater treatment system remains stable even under fluctuating substrate concentrations.

## Methods
### Experimental facilities
Two UASB reactors were operated in the experiment. The first UASB reactor, UASB1, was used as the control reactor, which operated without a sidestream unit. Another UASB reactors, UASB2, connected with a sidestream nitrite treatment unit, as shown in Fig. 1.

Two UASB reactors were made of plexiglass. The reaction zone was a cylinder with a height/diameter ratio of 10:1 and a reaction volume of 1 L. The UASB were covered with silver paper and black tape to avoid the growth of photosynthetic microorganisms. Influent was controlled by an influent pump, and synthetic wastewater was prepared every day. To maintain an anaerobic environment, nitrogen gas was used to reduce dissolved oxygen (DO) in synthetic wastewater.

When the DO was reduced to less than 1 mg·L$^{-1}$, the water surface was covered with plastic film to avoid oxygen dissolving into the synthetic wastewater. A small agitator, which connected the controller, was put into the reactor for mixing synthetic wastewater and sludge. A sampling port was set on the side of the reactor so a syringe could be used for quantitative sampling.

In the sidestream nitrite treatment unit, 100 ml mixture was taken from the UASB2 reactor into 400 ml of water (DO < 0.3 mg·L$^{-1}$), so the total volume of the sidestream unit was 500 ml. The sidestream unit was placed on a magnetic stirrer with a rotating speed of 100 rpm. Black plastic was used to cover the liquid surface to avoid oxygen dissolving during mixing, and DO and pH probes (WTW 3420, WTW Company Germany) were put into the sidestream unit for monitoring. Only nitrite was added to the sidestream unit. The maximum effluent nitrite concentration (30 mg·L$^{-1}$) of the UASB2 reactor was taken as the initial inhibition concentration. After nitrite treatment, the sludge was washed five times to ensure that the nitrite concentration was less than 1.0 mg·L$^{-1}$, and pumped back to the UASB2 reactor.

### Synthetic wastewater
NH$_4$Cl and NaNO$_2$ were used as substrates for anammox bacterial community, and the ratio of nitrite to ammonia in UASB influent was 1.0. Other synthetic wastewater components include (per liter): 450 mg NaHCO$_3$, 18 mg KH$_2$PO$_4$, 14 mg CaCl$_2$·2H$_2$O, 90 mg MgSO$_4$·7H$_2$O, and 1.25 mL trace elements. Trace element solution A contained (per liter): 5 g EDTA and 22 g FeSO$_4$, the Fe$^{2+}$ concentration remained at 0.08 mM during the whole experiment[81]. Trace element solution B contained (per liter): 15 g EDTA, 0.014 g H$_3$BO$_4$, 0.19 g NiCl$_2$·6H$_2$O, 0.22 g NaMoO$_4$·2H$_2$O, 0.25 g CuSO$_4$·5H$_2$O, 0.99 g MnCl$_2$·4H$_2$O, and 0.43 g ZnSO$_4$·7H$_2$O[82].

### Seed sludge
The seed sludge was 2.4 L of the floc anammox sludge which has been enriched in the laboratory. The sludge was evenly mixed and divided into two parts and inoculated into two UASB reactors. The mixed liquor suspended solids (MLSS) of each reactor were 6.516 g·L$^{-1}$. A biological sample (50 mL) was reserved for analysis of community structure.

### Operation of UASB reactors and sidestream nitrite treatment unit
The operation was divided into three phases.

In phase I (1–33 days), start-up of two anammox reactors. The initial hydraulic retention time (HRT) of two UASB reactors was 1.2 h for 23 days. Part of the sludge accumulated in the three-phase separator, which may be due to the high up-flow rate. To prevent sludge loss, the HRT of two reactors was adjusted to 2.4 h on day 24. When the effluent nitrite concentration was below 15 mg·L$^{-1}$, increasing the influent total nitrogen (TN) concentration to gradually increase the nitrogen loading rate (NLR) to meet the growing demands of anammox bacterial community.

In phase II (34–126 days), three nitrite exposure were performed. One-tenth of the sludge from UASB2 was exposed to nitrite in the sidestream unit. The sidestream unit was operated for ten days at a time, with a total of three operations during the whole phase, on the 34th, 68th and 117th days (see Supplementary Methods and Supplementary Table 1 for more details of nitrite treatment concentrations). UASB1 operated normally throughout this phase.

After phase II, to explore the adaptability of anammox bacterial community to high nitrite concentrations, sludge was taken out from the reactor to study the response of anammox bacterial community to different nitrite levels through the specific anammox activity (SAA) test[70]. The device used for the SAA test was the same as that of the sidestream unit. The sludge in each reactor was divided into five sections, with five separate devices. The NH$_4^+$-N concentration was fixed at

20 mg·L$^{-1}$, while the NO$_2$$^-$-N concentrations were 10, 20, 30, 40, 50 mg·L$^{-1}$, respectively. The medium was blown with N$_2$ to remove DO, and pH and temperature were measured. The MLSS was determined and the SAA was expressed as g·N·g·SS$^{-1}$·d$^{-1}$. After the SAA test, the sludge was starved at 4 °C for approximately two weeks.

In phase III (144–184 days), restart the two reactors. The sludge was returned to each UASB reactor and the temperature was restored to 31.0 ± 1.5 °C. To avoid substrate inhibition, influent TN concentration was increased gradually when the effluent nitrite concentration was below 15 mg·L$^{-1}$ to restore the nitrogen removal performance. On day 185, the nitrite shock test was operated by suddenly increasing the influent NO$_2$$^-$-N concentration from 80 mg·L$^{-1}$ to 200 mg·L$^{-1}$ to investigate the stability of two reactors, while the influent concentration of NH$_4$$^+$-N was stable at 80 mg·L$^{-1}$. The nitrite shock test was maintained for 5 h, the concentration of NH$_4$$^+$-N, NO$_2$$^-$-N, and NO$_3$$^-$-N in the reactor were detected every 30 min, and pH and temperature were monitored simultaneously. The decline rate of NRR was used to characterize the antifragility of the anammox reactor to high nitrite concentrations.

## Chemical analysis and calculations
The mixed liquor samples were taken by syringes and filtered through a 0.45 μm membrane immediately, and analysed for ammonium, nitrite, and nitrate with an Automatic multi-parameter water quality analyzer (DeChem-Tech Cleverchem Anna, Germany). The total nitrogen (TN) and nitrogen removal rate (NRR) was calculated accordingly. Free nitrous acid (FNA), the protonated form of nitrite, was be determined through the nitrite concentration, pH and temperature[83].

## Community structure analysis and visualization
Three sludge samples were collected. One samples from the initial moments of two UASB reactors (day 1) were named Seed. Two samples at the end of the two reactor operations (day 185) were named UASB1 and UASB2, respectively. Three replicates of amplicon sequencing were performed for each sample. DNA extraction and purification were conducted following the instruction of the E.Z.N.A.® soil DNA Kit (Omega Bio-tek, Norcross, GA, U.S.), and the DNA concentration was determined via using NanoDrop2000 (Thermo Scientific, Wilmington, USA). The primer pair of 338 F (5′-ACTCCTACGGGAGGCAGCA-3′) and 806 R (5′- GGACTACHVGGGTWTCTAAT-3′) were selected for polymerase chain reaction (PCR) amplification, and paired-end sequenced on the Illumina MiSeq PE300 platform (Illumina, San Diego, USA).

After quality filtering, a mean length ranging from 422 bp were obtained for each sample. To avoid unequal sequencing depth the number of valid sequences is 38223. These sequences were clustered into operational taxonomic units (OTUs) at a 97% similarity level. Based on the OTU clustering analysis, Mothur (v.1.30.2) software was used to analyze the OTU diversity index (alpha diversity analysis within samples)[84]. Beta diversity was calculated using QIIME, and hierarchical clustering was performed based on the beta diversity distance matrix[85]. To evaluate community assembly processes, the β-nearest taxon index (βNTI) and Bray-Curtis-based Raup-Crick (RC$_{Bray}$) were calculated through the package icamp (version 1.5.12)[86].

## Functional gene analysis
In order to understand the genes in the microbial community, all three samples were subjected to one metagenomic sequencing. Variation in the abundance of genes associated with metabolism in the anammox reaction was investigated by metagenomic sequencing, total genomic DNA was extracted from sludge samples using the E.Z.N.A.® Soil DNA Kit (Omega Bio-tek, Norcross, GA, U.S.) according to manufacturer's instructions. Concentration and purity of extracted DNA was determined with TBS-380 and NanoDrop2000, respectively. DNA extract quality was checked on 1% agarose gel. DNA extract was fragmented to an average size of about 400 bp for paired-end library construction.

Paired-end sequencing was performed according to the manufacturer's instructions. The raw data were trimmed (length< 50 bp or with a quality value < 20 or having N bases) by fastp (https://github.com/OpenGene/fastp, version 0.20.0). The size of metagenomic data of each sample ranges from 7.3 Gb to 8.8 Gb after quality control. The functional annotation was conducted using DIAMOND (version 0.8.35) against the Kyoto Encyclopedia of Genes and Genomes database (https://www.genome.jp/kegg/)[87,88].

## Reporting summary
Further information on research design is available in the Nature Portfolio Reporting Summary linked to this article.

## Data availability
All data generated in this study are provided in the article file, Supplementary Information, and Supplementary Data. The relevant source data from each figure are provided in the Source Data files. The amplicon sequence datasets generated during this study are available with accession number PRJNA1102518. The metagenomic sequence data could be found at BioProject PRJNA1094462. Source data are provided with this paper.

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

## Acknowledgements

This research was financially supported by the National Natural Science Foundation of China (52260003, U23A20675, to B.M.), the Hainan Provincial Natural Science Foundation of China (422QN274, to Y.W.) and the Key Research and Development Plan of Hainan Province (ZDYF2021SHFZ061, to B.M.).

## Author contributions

B.M., B.L., C.L., and Y.W. conceived the study. B.L. and C.L. performed the experiments, data analysis, and prepared the manuscript. B.M. supervised the whole research and revised the manuscript. J.B. performed the community assembly modeling. Y.H. visualized the data. R.S. revised the title of the manuscript. All authors are involved in the interpretation of the results and the preparation of this manuscript.

## Competing interests

The authors declare no competing interests.
