## [Peer Review File · Nature Communications]

Strategy to mitigate substrate inhibition in wastewater treatment systemsREVIEWER COMMENTS

Reviewer #1 (Remarks to the Author):

In their manuscript entitled "Enhancing microbial resistance: A novel approach to mitigate high-substrate inhibition in wastewater treatment systems" the authors describe a paired UASB experiment on microbial communities removing nitrogen via the anammox process. In one of the UASB reactors, they expose part of the microbial community to high nitrite concentration for 10 days and then return this subset of the community to the main UASB reactor. The authors then observe that the community is more resilient to subsequent nitrite accumulation. This is presented as a use case for the more general principle of antifragility of a microbial community to a stress, if the community has previously been exposed to a non-lethal version of this stressor. The authors pair their reactor data with 16S rRNA gene amplicon sequencing of the reactor seed community, and both end points of the experimental and control reactor. The amplicon sequencing shows strong community shifts in both the anammox organisms (Kuenenia to Jettenia) and the rest of the community (Denitratisoma to Nitrotoga).

Overall, I'm impressed by the reactor data in the manuscript, which clearly shows that the repeated nitrite stress improved the performance of UASB2 under high nitrite conditions. The reactor data presented is convincing that the sidestream unit substantially changes reactor performance. This is a nice result, with relevance for full scale applications of the anammox process in nitrogen removal systems.

It wasn't entirely clear to me how the sidestream unit was connected to the reactor during the 10 day periods it was in operation. From figure 1 it looks like that the effluent from the sidestream reactor is mixed with the influent from the main reactor, thus also exposing the main population in UASB2 to higher nitrite load. Is this correct? Please clarify and state this more clearly in the methods.

I have more doubts about the microbiology section of the manuscript. First, the authors frame the improved performance of their as enhancing the tolerance of anammox bacteria, but their sequencing data shows that a different anammox organism took over the community. In this sense, the community adapted, but not the organisms themselves. They got replaced by a different organism that is better adapted to a high nitrite environment. Several other members of the community also seem to be strongly affected by the exposure to high nitrite. Notably, Denitratisoma (likely denitrifiers) that are more abundant than the anammox organisms in the seed biomass and UASB1 are strongly reduced in UASB2, while Nitrotoga and SJA-28 increase in abundance. Nitrotoga is primarily known as a nitrite oxidizer, but in this anoxic system is unlikely to fill that niche. SJA-28 includes lineages OLB4 & OLB5 previously found in a partial nitrification anammox reactor in Olburgen, NL.

It would be nice to see how this community transition happened. Is there any frozen biomass from the reactor operation left that could be used for further amplicon sequencing? On a more conceptual note, as the authors framing of the study involves "antifragility", they should make more clear that it is the system itself (or the community in its entirety) that behaves antifragile. The anammox organisms originally most dominant seems to die off or wash out in response to the nitrite stress, so the organism themselves do not seem to gain increased tolerance and are not antifragile.

Most of section 3.4.2, which deals with the microbial community shifts, overinterprets the findings from the 16S amplicon survey. Many of the identified organisms are uncultured, and thus their metabolic or biosynthetic capabilities should be interpreted with some caution. For example, it is unclear whether OLB14, SJA-15 or SJA-28 produce more EPS than others in the community (line 133) and the order SJA-28 contains organisms with the capability for nitrate reduction, but also organisms that do not encode the genes for nitrate respiration (line 342-343). These are examples, but the trend holds for the entire section.

I suggest refocusing this section on the organisms undergoing the biggest changes between the treatments, such as Kuenenia/Jettenia, Denitratisoma, SJA-28 from fig 7, as well the organisms from figure S3 (notably nitrotoga), and either here or in the discussion put them into more comprehensive literature context. I suggest the authors speculate on the role of nitrotoga in this system, as the lack of oxygen should preclude a role in nitrite oxidation.

Regarding the literature cited for the microbiology section in the introduction and the results/discussion, please consider citing more primary literature or reviews, rather than recent papers supporting statements. As examples, citing a 2023 research paper for "nitrogen removal

involves nitrification and denitrification" or a 2018 research paper for "nitrification involves oxidation of ammonia to nitrate by nitrifying bacteria" is not the most appropriate way to cite. This again is a general point not limited to these two citations.

Please limit the use of tools that try to infer function from 16S rRNA gene amplicon sequencing (such as FAPROTAX and BugBase, but also picrust). In some cases, there is a clear link between taxonomy and function (eg anammox bacteria or oxygenic cyanobacteria) but there are many microbial traits that get lost or gained easily, and there is limited link between taxonomy and function. Especially in environments with poorly characterized organisms (such as this one) the use of these tools is inappropriate. I suggest removing section 3.4.3 and figure 8. Case in point is that the strongest effect seen in figure 8 is "nitrification/aerobic nitrite oxidation". If the reactors were indeed anaerobic (and I believe they were, otherwise there would have been ammonia oxidizing bacteria present), this process should not be happening, even though the increase of nitrotoga in the community drives this association.

Finally, I have some difficulties interpreting the microbial community figures in conjunction with the text. Is it possible that the labels in figure 6a were switched (ie the stacked bar labeled "seed" represents "UASB2")? Could it then also be that the hierarchical clustering dendrogram is mirrored? However, if that is the case, then I don't understand how the abundances in figure S3 can be right (since the total % of proteobacteria in fig 6 is lower than the % nitrotoga in figure S3). Or do those abundances in supplemental figure S3 only represent the % of the unique community? If so, please change those abundances to % in the total UASB community, so it is possible to assess their relevance for community functioning.

minor remarks:

line 19: The sentence ending with "pathogenic" 19 seems truncated

line 30: change "human" to "human sources"

line 42-43: There are more nitrifying bacteria (and archaea) than nitrobacter and nitrosomonas alone, I suggest just omitting the names instead of trying to provide a list.

line 46-48: not sure whether the PAO system is relevant for this manuscript, so I'd remove

line 79-80: it isn't the application of anammox bacteria that revealed their substrates, but the enrichment and physiology work. Please phrase and cite appropriately

line 195: OTU instead of OUT

line 411-417: I question the transferability of these results to the PAO system, since it doesn't seem like the microorganisms themselves gained increased tolerance, as is proposed in these lines.

the shading in figure two corresponds to FNA, which I assume is free nitrous acid, but this is not defined anywhere, nor is it stated how this is measured or calculated

the 3rd y axis in figure 3 has (%) as a unit, but I assume from the values that those are fractions

The statistical validation that the abundance of Jettenia (and the other taxa) is different between the reactors adds little to the manuscript, and I would move figure 7 to the supplement or remove it. Most of the information in figure 7 is also in figure 6, although the list of taxa isn't the same.

Reviewer #2 (Remarks to the Author):

This paper describes a strategy to improve the resilience of anammox reactors to perturbations, which is an important area of research in the application of anammox for real-world wastewater treatment. In practice, the data suggests that inoculum sludge could be better selected for full-scale plants based on SAA tests, which is a useful practical application of the research.

My main criticism of the paper is that it does not, in fact, describe a strategy to make particular anammox bacteria more resistant, and there is no data to support that claim in the paper. Instead, the paper demonstrates that different cultivation conditions will select for different populations within communities that are better adapted to those cultivation conditions. In this context, the lack of reactor replication to support the community analysis is a weakness of the paper.

Some examples:

The title of the paper is a bit misleading, in the sense that the outcome was not a more "resistant" microbe, but rather a more resistant system underpinned by ecological shift towards organisms more comfortable at higher nitrite levels.

Results line 212: "The results showed that the treatment with non-lethal high nitrite environment has enhanced the tolerance of anammox bacteria". Is it more correct to state that the treatment has selected for different, more tolerant, anammox bacteria? The dominant anammox bacteria are indeed different in both reactors "This non-lethal high nitrite environment changed the dominant anammox bacteria from *Candidatus Kuenenia* to *Candidatus Jettenia*". This is perhaps not surprising given the different cultivation conditions applied in each reactor and it is well known, and acknowledged by the authors, that "*Candidatus Jettenia* is often the dominant anammox population in response to increased substrate load, as a high nitrogen concentration may be more favorable for *Candidatus Jettenia*".

Results line 293: "This non-lethal exposure and timely recovery of anammox bacteria likely acquired a heightened resistance and improved nitrite tolerance, which potentially enhanced the ability to withstand and adapt to varying nitrite levels." This statement is not correct as there is no evidence that any organisms acquired a heightened resistance. Rather, there is evidence that the nitrite treatments selected for different, more tolerant anammox organisms. This conclusion is also supported by the better reactor responses to the shocks over time as the community developed.

My other main comment relates to the molecular analysis:

Results line 304: the Beta diversity analysis and hierarchical clustering analysis underlined the differences between the communities in both reactors - "The analysis revealed that the UASB2 sludge samples were dissimilar from the other samples based on phylogeny."

These differences were not restricted to the anammox bacteria, and indeed the paper also later (line 314) reports a significant decline in the Proteobacteria and other major shifts in community composition at a phylum level. The authors present an analysis of the sequence data to assess the impact of the nitrite treatments on other functional groups; and on the factors driving community development in UASB2. This is a strength of the paper as it offers the opportunity to examine community development in response to the different cultivation conditions. I think the sampling strategy may not have been focused on this wider aspect, given the lack of reactor replication, or at least the outcome of any community assembly modelling and other tools were not presented. The application of such tools, if appropriate to the sampling plan and sample numbers, would strengthen the paper.

Minor comments:

Page 7 line 169: "In phase III (144–184 days), the sludge was returned to each UASB reactor and the temperature was restored to $31.0 \pm 1.5^\circ\text{C}$ " – had the temperature been changed before this time?

Abstract line 18: "reducing potentially pathogenic" here doesn't fit with the topic of the paper

Detailed response to reviewers' comments

We would like to thank all reviewers for their valuable comments. We have carefully revised the manuscript considering all these comments. Below are our detailed responses to the reviewer's comments, according to the Editor instructions.

Reviewer #1 (Remarks to the Author):

In their manuscript entitled "Enhancing microbial resistance: A novel approach to mitigate high-substrate inhibition in wastewater treatment systems" the authors describe a paired UASB experiment on microbial communities removing nitrogen via the anammox process. In one of the UASB reactors, they expose part of the microbial community to high nitrite concentration for 10 days and then return this subset of the community to the main UASB reactor. The authors then observe that the community is more resilient to subsequent nitrite accumulation. This is presented as a use case for the more general principle of antifragility of a microbial community to a stress, if the community has previously been exposed to a non-lethal version of this stressor.

The authors pair their reactor data with 16S rRNA gene amplicon sequencing of the reactor seed community, and both end points of the experimental and control reactor.

*The amplicon sequencing shows strong community shifts in both the anammox organisms (*Kuenenia* to *Jettenia*) and the rest of the community (*Denitratisoma* to *Nitrotoga*).*

Overall, I'm impressed by the reactor data in the manuscript, which clearly shows that the repeated nitrite stress improved the performance of UASB2 under high nitrite

*conditions. The reactor data presented is convincing that the sidestream unit*
*substantially changes reactor performance. This is a nice result, with relevance for*
*full scale applications of the anammox process in nitrogen removal systems.*

Response:

Thank you for your positive evaluation of our manuscript. We are pleased to hear
that our UASB experiment and data have impressed you, especially the evident
improvement in the performance of UASB2 under high nitrite conditions after
repeated nitrite stress. Your recognition of the relevance of our findings for
full-scale applications of the anammox process in nitrogen removal systems is
very encouraging. We have carefully considered and adopted your precious
comments and suggestions for improvement in the revision of our manuscript.

1) *It wasn't entirely clear to me how the sidestream unit was connected to the*
*reactor during the 10 day periods it was in operation. From figure 1 it looks like that*
*the effluent from the sidestream reactor is mixed with the influent from the main*
*reactor, thus also exposing the main population in UASB2 to higher nitrite load. Is*
*this correct? Please clarify and state this more clearly in the methods.*

Response:

Thank you for your questions regarding the connection.

In fact, we operated the sidestream unit in such a way as to minimise the impact
of high nitrite concentrations on the main reactor. Specifically, during the ten
44 days that the sidestream unit was in operation, 100 ml of sludge mixture was

45 transferred daily from the UASB2 reactor, and mixed with 400 ml of purified
water. And then injected high concentration of nitrite solution (0.375-1.25 mL,
40 g/L) to achieve the set treatment concentration. Following the treatment, the
sludge was washed five times under low dissolved oxygen conditions. After
detecting that the nitrite concentration was below 1 mg/L, the settled sludge
(about 70 ml) was then returned to the UASB2 reactor. Throughout this operation,
we avoided bringing high nitrite loads into the reactor, achieving that the
bacterial community was only exposed to high nitrite in the sidestream unit,
while avoiding other environmental conditions (e.g., dissolved oxygen) from
affecting the anammox sludge.

To prevent confusion, we have revised the Methods section to include the
following details, and also updated Fig. 1:

“After nitrite treatment, the sludge was washed five times to ensure that the nitrite
concentration was less than $1.0 \text{ mg}\cdot\text{L}^{-1}$, and pumped back to the UASB2 reactor.” (Page 5,
Line 124-126)

"Fig. 1. Device details of two UASB reactors. UASB1 was the control reactor, UASB2
was equipped with a sidestream unit separately." (Page 27, Line 776-778)

2) *I have more doubts about the microbiology section of the manuscript. First, the*
*authors frame the improved performance of their as enhancing the tolerance of*
*anammox bacteria, but their sequencing data shows that a different anammox*
*organism took over the community. In this sense, the community adapted, but not the*
*organisms themselves. They got replaced by a different organism that is better*
*adapted to a high nitrite environment. Several other members of the community also*
*seem to be strongly affected by the exposure to high nitrite. Notably, Denitratisoma*
*(likely denitrifiers) that are more abundant than the anammox organisms in the seed*
*biomass and UASB1 are strongly reduced in UASB2, while Nitrotoga and SJA-28*
*increase in abundance. Nitrotoga is primarily known as a nitrite oxidizer, but in this*
*anoxic system is unlikely to fill that niche. SJA-28 includes lineages OLB4 & OLB5*
*previously found in a partial nitrification anammox reactor in Olburgen, NL.*
*It would be nice to see how this community transition happened. Is there any frozen*
*biomass from the reactor operation left that could be used for further amplicon*
*sequencing? On a more conceptual note, as the authors framing of the study involves*
*"antifragility", they should make more clear that it is the system itself (or the*
*community in its entirety) that behaves antifragile. The anammox organisms*
*originally most dominant seems to die off or wash out in response to the nitrite stress,*
*so the organism themselves do not seem to gain increased tolerance and are not*
*antifragile.*

Response:

We appreciate your insights into the adaptive aspects of microbial communities.

We agree with your observation that the microbial community, rather than the
organisms themselves, adapted to the high nitrite environment. In order to avoid
confusion for the reader, we have changed the term "anammox bacteria" to
"anammox bacterial community" in the manuscript. And we use the term
"antifragile" in the paper and ensure that it was consistent with system.

Other microbial community also showed response to high nitrite environments,
such as the denitrifying bacterial community. *Denitratisoma* and *SJA-28* both
have the ability to reduce nitrate, a by-product of anammox process, to nitrite.
The increase of *SJA-28* in its abundance in UASB2 could be attributed to its
ability to utilize nitrite as an electron acceptor in denitrification, which would
provide a competitive advantage in an environment with high nitrite
concentrations, such as the sidestream unit of UASB2 system.

We sincerely apologize for the confusion caused by the abundance of *Nitrotoga*
in UASB2. This confusion arises from the calculation method used in Figure S3,
where the abundance of *Nitrotoga* was presented as a percentage of the 20
selected genera, rather than as a percentage of the total microbial community. To
clarify, *Nitrotoga* is indeed very underrepresented in UASB2, with an abundance
of only 0.0594% of the total microbial community. This low abundance was not
accurately reflected in the original presentation of Figure S3, leading to your
perception of a higher abundance. To address this issue, we have revised Figure
S3 to correctly represent the abundance of *Nitrotoga* as a percentage of the total
UASB2 community.

Fortunately, we had frozen samples preserved. Upon receiving your feedback, we
resequenced each frozen sample three times and performed community assembly
modelling based on the updated data. The results of this analysis indicate that,
compared to UASB1, heterogeneous selection dominated for microbial
community in UASB2, nitrite concentration was speculated to be the main
driving factor in our study. We have included our methodological details and
results of community assembly modelling in the revised manuscript, as follows:

"Beta diversity was calculated using QIIME, and hierarchical clustering was performed
based on the beta diversity distance matrix. To evaluate community assembly processes,
the β -nearest taxon index (β NTI) and Bray-Curtis-based Raup-Crick (RCBray) were
calculated through the package icamp (version1.5.12) ⁵¹." (Page 9, Line 201-205)

"The variation in the community assembly process was further investigated by comparing
the β NTI index, and the results of the β NTI analyses were used to calculate the
proportions of deterministic and stochastic processes (Fig.5c). The results showed that
heterogeneous selection dominated for microbial communities in UASB2, with basic
abiotic parameters as the driving factors ⁵⁵. Nitrite concentration was probably the main
driver in this study." (Page 13, Line 324-329)

"Fig. 5. (a) Species richness expressed via Ace and Chao estimator; (b) Species evenness expressed via the Shannon and Simpson indices; (c) The summary of assembly processes." (Page 31, Line 791-794)

"Figure S2. (a) The number of unique and shared genera identified across the three different sample; (b) the abundance of Unique genera in UASB2." (Supplementary materials)

3) *Most of section 3.4.2, which deals with the microbial community shifts,*
*overinterprets the findings from the 16S amplicon survey. Many of the identified*
*organisms are uncultured, and thus their metabolic or biosynthetic capabilities*
*should be interpreted with some caution. For example, it is unclear whether OLB14,*
*SJA-15 or SJA-28 produce more EPS than others in the community (line 133) and the*
*order SJA-28 contains organisms with the capability for nitrate reduction, but also*
*organisms that do not encode the genes for nitrate respiration (line 342-343). These*
*are examples, but the trend holds for the entire section.*

*I suggest refocusing this section on the organisms undergoing the biggest changes*
*between the treatments, such as Kuenenia/Jettenia, Denitratisoma, SJA-28 from fig 7,*
*as well the organisms from figure S3 (notably nitrotoga), and either here or in the*
*discussion put them into more comprehensive literature context. I suggest the authors*
*speculate on the role of nitrotoga in this system, as the lack of oxygen should preclude*
*a role in nitrite oxidation.*

Response:

We thank you for your suggestions of the microbial community change section.

We agree with you, and in the revision process, we have removed the discussion
and analysis of other low abundance genera to avoid over-interpreting the
metabolic or biosynthetic capabilities of uncultured microbes. We have extended
the analysis on the organisms undergoing the biggest changes, such as
*Kuenenia/Jettenia, Denitratisoma, SJA-28*, to better understand the role of these
genera in the system.

In light of your suggestion, we have refrained from further analysis of *Nitrotoga*
due to its very low abundance, as evidenced by the revised Figure S3, where the
abundance of *Nitrotoga* was correctly represented as 0.0594% of the total
microbial community.

The revised 3.4.2 was shown as follows:

"3.4.2 High-nitrite environments shift the structure of microbial communities

The study found that the abundance of three phyla, *Planctomycetes*, *Chloroflexi* and
*Bacteroidetes*, increased in UASB2 as compared to UASB1, while there was a significant
decrease in the abundance of *Proteobacteria* (Fig.6a). *Planctomycetes* phylum contained
three types anammox genera, *Candidatus* Kuenenia, *Candidatus* Brocadia and
*Candidatus* Jettenia (Fig.6b). Among these, *Candidatus* Kuenenia was the dominant
genera in the seed sludge and UASB1 sludge, the relative abundance was 9.93% and
12.62%, respectively. The dominance of *Candidatus* Kuenenia can be attributed to its
k-strategist characteristics, making it more competitive under conditions of low nitrogen
concentrations ^{56, 57, 58}.

After nitrite exposure, the main anammox bacterial genera were *Candidatus* Jettenia,
accounting for 28.26% in UASB2, while the remaining two genera accounted for less
than 0.5%. *Candidatus* Kuenenia was decreased abruptly due to their susceptibility to
fluctuating substrate conditions ⁵⁹. This transformation in response to nitrite shock
potentially contributed to the antifragility of UASB2 reactor. Since *Candidatus* Jettenia
may possess superior adaptive advantages in high nitrite concentrations compared to other
anammox genera, it was able to adapt well to influent disturbance. The variation of

anammox bacterial community in UASB2 was consistent with the diversity. The adapted
groups grew and multiplied, and the non-adapted groups withered and died, changing the
proportions of different genera in the anammox bacterial community.

*Denitratisoma* and *norank_c__SJA-28* are the two genera in the UASB2 system with
relatively significant abundance variations, except anammox genera. Both *Denitratisoma*
and *norank_c__SJA-28* have the ability to participate in denitrification reactions^{60, 61, 62},
which was important in anammox systems^{63, 64}. Nevertheless, *Denitratisoma* decreased in
abundance at high nitrite concentrations⁶⁵. In contrast, the *SJA-28* lineage was able to
encode an ammonia-forming cytochrome c nitrite reductase of the NrfAH-type involved
in nitrite reduction⁶⁶. That may allow *SJA-28* survive higher nitrite concentrations, which
may explain the increased abundance in UASB2.

Differences of bacterial communities in UASB2 may be due to different responses to
nitrite-induced stress. Further exploration of microbial functions could provide insights
into changes in community distribution." (Page 14, Line 337-368)

4) *Regarding the literature cited for the microbiology section in the introduction*
*and the results/discussion, please consider citing more primary literature or reviews,*
*rather than recent papers supporting statements. As examples, citing a 2023 research*
*paper for "nitrogen removal involves nitrification and denitrification" or a 2018*
*research paper for "nitrification involves oxidation of ammonia to nitrate by*
*nitrifying bacteria" is not the most appropriate way to cite. This again is a general*
*point not limited to these two citations.*

Response:

Thank you for the comments on the literature citations. We have reviewed and
updated about twenty citations throughout the manuscript. This modification
provides a stronger foundation for our argument and improves the overall
credibility of the cited literature. The following are examples of the revisions:

"The enrichment and physiology work demonstrated that nitrite is an important substrate
for anammox bacteria, but nitrite with a high level can also act as an inhibitor 44, 45."

(Page 3, Line 76-78)

44. Strous M, Kuenen JG, Jetten MS. Key physiology of anaerobic ammonium
oxidation. *Applied and environmental microbiology* 65, 3248-3250 (1999).

45. van de Graaf AA, de Bruijn P, Robertson LA, Jetten MSM, Kuenen JG.
Autotrophic growth of anaerobic ammonium-oxidizing micro-organisms in a fluidized
bed reactor. *Microbiology* 142, 2187-2196 (1996).

5) *Please limit the use of tools that try to infer function from 16S rRNA gene*
*amplicon sequencing (such as FAPROTAX and BugBase, but also picrust). In some*
*cases, there is a clear link between taxonomy and function (eg anammox bacteria or*
*oxygenic cyanobacteria) but there are many microbial traits that get lost or gained*
*easily, and there is limited link between taxonomy and function. Especially in*
*environments with poorly characterized organisms (such as this one) the use of these*
*tools is inappropriate. I suggest removing section 3.4.3 and figure 8.*

*Case in point is that the strongest effect seen in figure 8 is "nitrification/aerobic*

*nitrite oxidation". If the reactors were indeed anaerobic (and I believe they were,*
*otherwise there would have been ammonia oxidizing bacteria present), this process*
*should not be happening, even though the increase of nitrotoga in the community*
*drives this association.*

Response:

Thank you very much for your feedback on the use of functional inference tools.
We have removed Section 3.4.3 and Figure 8 from the revised manuscript to
address your concerns.

In response to your suggestion, we conducted metagenomic sequencing on the
frozen samples to gain a deeper understanding of the key functional genes of the
anammox bacterial communities. We have included our methodological details
and results in the revised manuscript, as follows:

"2.7. Functional gene analysis

Variation in the abundance of genes associated with metabolism in the anammox reaction
investigated by metagenomic sequencing, total genomic DNA was extracted from sludge
samples using the E.Z.N.A.® Soil DNA Kit (Omega Bio-tek, Norcross, GA, U.S.)
according to manufacturer's instructions. Concentration and purity of extracted DNA was
determined with TBS-380 and NanoDrop2000, respectively. DNA extract quality was
checked on 1% agarose gel. DNA extract was fragmented to an average size of about 400
bp for paired-end library construction. Paired-end sequencing was performed according to
the manufacturer's instructions. The raw data were trimmed (length<50 bp or with a
quality value <20 or having N bases) by fastp (<https://github.com/OpenGene/fastp>,

version 0.20.0). The functional annotation was conducted using Diamond (version 0.8.35)
against the Kyoto Encyclopedia of Genes and Genomes database
(<http://www.genome.jp/keeg/>)." (Page 9, Line 207-219)

"3.4.3 High-nitrite environment transformed the community contribution to functional
genes

The anammox process involves the conversion of nitrite and ammonia to N₂ gas through
three possible pathways. This conversion process relies on the synergistic action of three
key functional enzymes: hydrazine synthase (HZS), hydrazine dehydrogenase (HDH),
and hydroxylamine oxidoreductase (HAO) ⁶⁷. HZS catalyses the conversion of NO from
nitrite to hydrazine (N₂H₄) with NH₄⁺. The N₂H₄ generated is then oxidized by HDH and
eventually converted to N₂ ⁶⁸. The formation of hydrazine may also occur through the
intermediate of hydroxylamine (NH₂OH), which can be converted to NO by HAO,
participating in the anammox process ⁶⁹.

The genes producing these enzymes were detected by metagenomic analysis to
investigate the contribution of anammox bacterial community to key functional genes. It
was found that the functional genes *hzs*, *hdh* and *hao*, encoded by *Candidatus* Kuenenia
in the UASB1, were highly abundant, contributing 91.76%, 92.37%, and 80.82%,
respectively (Fig.7). After the nitrite treatment, the abundance of functional genes
encoded by *Candidatus* Jettenia increased significantly and became the largest
contributing anammox genus. It indicated that the anammox process was dominated by
*Candidatus* Jettenia bacteria in UASB2, which was consistent with the change in
community structure." (Page 15, Line 370-388)

"Fig. 7. The contribution of anammox bacterial communities to key functional genes, the

left side was untreated with nitrite, the right side was treated with nitrite." (Page 33, Line

798-800)

6) Finally, I have some difficulties interpreting the microbial community figures in

conjunction with the text. Is it possible that the labels in figure 6a were switched (ie

the stacked bar labeled "seed" represents "UASB2")? Could it then also be that the

hierarchical clustering dendrogram is mirrored? However, if that is the case, then I

don't understand how the abundances in figure S3 can be right (since the total % of

proteobacteria in fig 6 is lower than the % nitrotoga in figure S3). Or do those

abundances in supplemental figure S3 only represent the % of the unique community?

If so, please change those abundances to % in the total UASB community, so it is

possible to assess their relevance for community functioning.

Response:

We apologize for these embarrassing errors in Figure 6. An error has been made

in the labelling of the stacked bar plot in Figure 6a. The stacked bar labelled

"Seed" should represent "UASB2" and vice versa. We have corrected this error.
 The low abundance was not accurately reflected in the original presentation of
 Figure S3, and we have revised Figure S3 to correctly represent the abundance of
 *Nitrotoga* as a percentage of the total UASB2 community.
 The revised figures were as follows:

 "Fig. 6. (a) Hierarchical clustering tree and distribution of the bacterial community on
 phylum level. (b) Analysis of bacterial community abundance at the genus level." (Page
 32, Line 795-797)

 "Figure S2. (a) The number of unique and shared genera identified across the three
 different sample; (b) the abundance of Unique genera in UASB2." (Supplementary
 materials)

**minor remarks:**

1) *line 19: The sentence ending with "pathogenic" 19 seems truncated*

Response:

We appreciate your observation regarding the truncated sentence on line 19. As
301 per your earlier suggestion, we have removed the section of the manuscript that
relied on 16S rRNA data for function prediction. Consequently, the sentence in
question, which referenced the BugBase predictions, has been deleted from the
abstract.

2) *line 30: change "human" to "human sources"*

Response:

The term “human” has been revised to “human sources” (line 30).

3) *line 42-43: There are more nitrifying bacteria (and archaea) than nitrobacter*
*and nitrosomonas alone, I suggest just omitting the names instead of trying to provide*
*a list.*

Response:

We have revised the manuscript to omit the specific names of nitrifying bacteria
and archaea. The revised sentence is as follows:

“The process of nitrification involves the oxidation of ammonia to nitrate by
ammonia-oxidizing bacteria and nitrite-oxidizing bacteria” (Page 2, Line 38-40)

4) *line 46-48: not sure whather the PAO system is relevant for this manuscript, so*
*I'd remove.*

Response:

We have removed the section about PAOs system.

5) *line 79-80: it isn't the application of anammox bacteria that revealed their*
*substrates, but the enrichment and physiology work. Please phrase and cite*
*appropriately*

Response:

We agree with you that it was not through the direct application of these bacteria
that their substrates were revealed, but rather through enrichment and physiology
work.

This sentence has been revised as follows:

“The enrichment and physiology work demonstrated that nitrite is an important substrate
for anammox bacterial communities, but excessive nitrite concentrations can also act as
an inhibitor^{44, 45}.” (Page 4, Line 76-78)

44. Strous M, Kuenen JG, Jetten MS. Key physiology of anaerobic ammonium
oxidation. *Applied and environmental microbiology* 65, 3248-3250 (1999).

45. van de Graaf AA, de Bruijn P, Robertson LA, Jetten MSM, Kuenen JG.
Autotrophic growth of anaerobic ammonium-oxidizing micro-organisms in a fluidized
bed reactor. *Microbiology* 142, 2187-2196 (1996).

6) *line 195: OTU instead of OUT*

Response:

The term “OUT” has been revised to “OTU”.

7) *line 411-417: I question the transferability of these results to the PAO system,*

*since it doesn't seem like the microorganisms themselves gained increased tolerance,*

*as is proposed in these lines.*

Response:

We appreciate your question as to whether the results of this paper can be applied

to other systems.

This paper has confirmed that the non-lethal exposure to nitrite in the side-stream

unit can enhance the tolerance of anammox bacterial communities to high

concentrations of nitrite. The applicability of this conclusion to microbial

communities in wastewater treatment system depends on the presence of genera

capable of tolerating high substrate concentrations.

For instance, in the case of PAOs, *Candidatus Accumulibacter* Clade IIC was

found to be sensitive to nitrite, whereas Clade IID exhibited strong tolerance to

nitrite exposure. Therefore, this conclusion could be applied to PAOs. Under

high nitrite levels (above 16 mg/L), anoxic phosphorus uptake was primarily

carried out by Clade IID due to its strong tolerance to nitrite exposure. Therefore,

the conclusion of this study is considered applicable to PAOs.

In addition, we have also included in the revised manuscript an introduction to
the tolerance of different microbial genera to inhibition by high concentrations of
ammonia and sulfide. This further demonstrates the feasibility of our approach
across various microbial systems.

The specific revisions to the manuscript are as follows:

“Creating non-lethal high substrate inhibition environment could be used to improve
resistance not only to nitrite, but also has the potential to be used for other substrates, such
as ammonia and sulfur are present in wastewater treatment processes^{79,80}. Variations in
microbial tolerance to different substrate inhibition in wastewater treatment have been
investigated extensively. For instance, *Nitrosomonas* cluster 7 has a stronger tolerance to
high ammonia than other ammonia oxidizing bacteria⁸¹, sulphur-oxidising bacteria
*Chromatiaceae* are more tolerant of sulfur (thiosulphate) than *Thiobacillus*⁸², *Ca.*
*Accumulibacter* clades IID involved in denitrifying phosphorus removal via nitrite
pathway showed strong tolerance to nitrite than clades IIC⁸³. Consequently, the
non-lethal high substrate could use to increase tolerance of functional microbial
communities. The creation of a non-lethal high substrate environment in this study was
achieved using the sidestream treatment unit. This may be a feasible strategy to enhance
the antifragility of the system by improving the tolerance of the microbial community
without affecting normal operation. It is a potential approach to ensure that the
performance of the wastewater treatment system remains stable even under fluctuating
substrate concentrations.” (Page 18, Line 426-441)

8) The shading in figure two corresponds to FNA, which I assume is free nitrous
 acid, but this is not defined anywhere, nor is it stated how this is measured or
 calculated

Response:

We appreciate your attention to detail and have addressed the issue in the
 Materials and Methods section, as follows:

“Free nitrous acid (FNA), the protonated form of nitrite, was determined through the
 nitrite concentration, pH and temperature ⁵⁰” (Page 8, Line 186-187)

9) The 3rd y axis in figure 3 has (%) as a unit, but I assume from the values that
 those are fractions

Response:

We appreciate your observation and agree that the term “TN removal rate (%)” in
 Figure 3 could be misleading. We have revised the figure to use “TN removal
 efficiency (%)” instead, which more accurately reflects the fraction or percentage
 of total nitrogen that is removed.

“Fig. 3. The nitrogen removal performance of effluent nitrite concentrations, NRR and
 TN removal efficiency under nitrite shock. (a) was UASB1 reactor and (b) was UASB2
 reactor.” (Page 29, Line 783-786)

*10) The statistical validation that the abundance of Jettenia (and the other taxa) is*
*different between the reactors adds little to the manuscript, and I would move figure 7*
*to the supplement or remove it. Most of the information in figure 7 is also in figure 6,*
*although the list of taxa isn't the same.*

Response:

Thank you for your suggestion, and we have removed figure 7.

**Reviewer #2 (Remarks to the Author):**

*This paper describes a strategy to improve the resilience of anammox reactors to*
*perturbations, which is an important area of research in the application of anammox*
*for real-world wastewater treatment. In practice, the data suggests that inoculum*
*sludge could be better selected for full-scale plants based on SAA tests, which is a*
*useful practical application of the research.*

Response:

We are grateful for your positive assessment of our research, which focuses on
improving the resistance of anammox reactors to perturbations. We agree that the
practical application of our findings, such as using SAA tests to select inoculum
sludge for full-scale plants, is a valuable contribution to the field. Thank you very
much for your encouraging feedback.

1) *My main criticism of the paper is that it does not, in fact, describe a strategy to*
*make particular anammox bacteria more resistant, and there is no data to support*
*that claim in the paper. Instead, the paper demonstrates that different cultivation*
*conditions will select for different populations within communities that are better*
*adapted to those cultivation conditions. In this context, the lack of reactor replication*
*to support the community analysis is a weakness of the paper.*

Response:

Thanks for your valuable comments. We agree with your understanding. We have
changed "anammox bacteria" to "anammox bacterial community" in the
manuscript to provide readers with an accurate understanding.

We appreciate your comment regarding the lack of reactor replication to support
community analysis. We have addressed this limitation by repeating
high-throughput sequencing three times for each sample, and we have performed
community assembly modelling based on the data. At the same time, we added
metagenomic sequencing to investigate the contribution of anammox bacterial
communities to key functional genes.

2) *The title of the paper is a bit misleading, in the sense that the outcome was not a*
*more “resistant” microbe, but rather a more resistant system underpinned by*
*ecological shift towards organisms more comfortable at higher nitrite levels.*

Response:

We appreciate your insightful comment about the title. We agree that the title may
be misleading. The increased antifragility of the system in our results was
attributed to a shift in genera, making the microbial community more adaptable
to higher nitrite levels.

To address this, we propose the following revised title for our paper:

“Enhancing System Antifragility: A Novel Approach to Mitigate High-Substrate
Inhibition in Wastewater Treatment Systems” (Page 1, Line 1-2)

3) *Results line 212: “The results showed that the treatment with non-lethal high*
*nitrite environment has enhanced the tolerance of anammox bacteria”. Is it more*
*correct to state that the treatment has selected for different, more tolerant, anammox*

*bacteria? The dominant anammox bacteria are indeed different in both reactors*
*“This non-lethal high nitrite environment changed the dominant anammox bacteria*
*from Candidatus Kuenenia to Candidatus Jettenia”.* *This is perhaps not surprising*
*given the different cultivation conditions applied in each reactor and it is well known,*
*and acknowledged by the authors, that “Candidatus Jettenia is often the dominant*
*anammox population in response to increased substrate load, as a high nitrogen*
*concentration may be more favorable for Candidatus Jettenia”.*

Response:

We appreciate your comments and fully agree with your observation that the
microbial community, rather than the organisms themselves, adapted to the high
nitrite environment. Consequently, we have changed the term “anammox bacteria”
to “anammox bacterial community” in this sentence, and updated the entire
manuscript-

*“These results indicate that anammox bacterial community in UASB2 were more tolerant*
*to high nitrite levels.” (Page 10, Line 241-242)*

4) *Results line 293: “This non-lethal exposure and timely recovery of anammox*
*bacteria likely acquired a heightened resistance and improved nitrite tolerance,*
*which potentially enhanced the ability to withstand and adapt to varying nitrite levels.”*
*This statement is not correct as there is no evidence that any organisms acquired a*
*heightened resistance. Rather, there is evidence that the nitrite treatments selected for*
*different, more tolerant anammox organisms. This conclusion is also supported by the*
*better reactor responses to the shocks over time as the community developed.”*

Response:

We agree with you that there is no evidence that any organism has acquired
enhanced resistance. We believe that the microbial community has adapted to the
high nitrite environment. Hence, we have revised the term "anammox bacteria" to
"anammox bacterial community" in this sentence. And we have updated the
entire manuscript, in order to accurately reflect this interpretation of our results:

“The exposure of the anammox bacterial community to high nitrite concentrations in the
sidestream unit played a crucial role in enhancing their nitrite tolerance. Upon returning to
the normally operating UASB2 reactor, the anammox bacterial community had the
opportunity to recover under low nitrite concentrations and simultaneous presence of
ammonia. This non-lethal exposure and timely recovery of anammox bacterial community
likely acquired a heightened resistance and improved nitrite tolerance, which potentially
enhanced the ability to withstand and adapt to varying nitrite levels.” (Page 13, Line
302-309)

5) *My other main comment relates to the molecular analysis:*

*Results line 304: the Beta diversity analysis and hierarchical clustering analysis*
*underlined the differences between the communities in both reactors - “The analysis*
*revealed that the UASB2 sludge samples were dissimilar from the other samples*
*based on phylogeny.*

*These differences were not restricted to the anammox bacteria, and indeed the paper*
*also later (line 314) reports a significant decline in the Proteobacteria and other*

*major shifts in community composition at a phylum level. The authors present an*
*analysis of the sequence data to assess the impact of the nitrite treatments on other*
*functional groups; and on the factors driving community development in UASB2. This*
*is a strength of the paper as it offers the opportunity to examine community*
*development in response to the different cultivation conditions. I think the sampling*
*strategy may not have been focused on this wider aspect, given the lack of reactor*
*replication, or at least the outcome of any community assembly modelling and other*
*tools were not presented. The application of such tools, if appropriate to the sampling*
*plan and sample numbers, would strengthen the paper.*

Response:

We appreciate your comments on the molecular analysis. We very much agree
with you about one of the strengths of this paper. We did not sample and analyse
enough, but fortunately we froze the samples. Upon receiving your feedback, we
resequenced each frozen sample three times and performed community assembly
modelling based on the updated data. Concurrently, we conducted metagenomic
sequencing on the sludge samples to investigate the contribution of anammox
bacterial communities to key functional genes.

We have included our methodological details and results in the revised
manuscript, as follows:

"Beta diversity was calculated using QIIME, and hierarchical clustering was performed
based on the beta diversity distance matrix. To evaluate community assembly processes,
the β -nearest taxon index (β NTI) and Bray-Curtis-based Raup-Crick (RCBray) were
calculated through the package icamp (version1.5.12)⁵¹." (Page 9, Line 201-205)

"The variation in the community assembly process was further investigated by comparing
the β NTI index, and the results of the β NTI analyses were used to calculate the
proportions of deterministic and stochastic processes (Fig.5c). The results showed that
heterogeneous selection dominated for microbial communities in UASB2, with basic
abiotic parameters as the driving factors 55. Nitrite concentration was probably the main
driver in this study." (Page 13, Line 324-329)

"2.7. Functional gene analysis

Variation in the abundance of genes associated with metabolism in the anammox reaction
investigated by metagenomic sequencing, total genomic DNA was extracted from sludge
samples using the E.Z.N.A.® Soil DNA Kit (Omega Bio-tek, Norcross, GA, U.S.)
according to manufacturer's instructions. Concentration and purity of extracted DNA was
determined with TBS-380 and NanoDrop2000, respectively. DNA extract quality was
checked on 1% agarose gel. DNA extract was fragmented to an average size of about 400
537 bp for paired-end library construction. Paired-end sequencing was performed according to
538 the manufacturer's instructions. The raw data were trimmed (length<50 bp or with a
539 quality value <20 or having N bases) by fastp (<https://github.com/OpenGene/fastp>,
version 0.20.0). The functional annotation was conducted using Diamond (version 0.8.35)
against the Kyoto Encyclopedia of Genes and Genomes database
(<http://www.genome.jp/keeg/>)." (Page 9, Line 207-219)

"3.4.3 High-nitrite environment transformed the community contribution to functional
genes

The anammox process involves the conversion of nitrite and ammonia to N₂ gas through
three possible pathways. This conversion process relies on the synergistic action of three
key functional enzymes: hydrazine synthase (HZS), hydrazine dehydrogenase (HDH),
and hydroxylamine oxidoreductase (HAO) ⁶⁷. HZS catalyses the conversion of NO from
nitrite to hydrazine (N₂H₄) with NH₄⁺. The N₂H₄ generated is then oxidized by HDH and
eventually converted to N₂ ⁶⁸. The formation of hydrazine may also occur through the
intermediate of hydroxylamine (NH₂OH), which can be converted to NO by HAO,
participating in the anammox process ⁶⁹.

The genes producing these enzymes were detected by metagenomic analysis to
investigate the contribution of anammox bacterial community to key functional genes. It
was found that the functional genes *hzs*, *hdh* and *hao*, encoded by *Candidatus* Kuenenia
in the UASB1, were highly abundant, contributing 91.76%, 92.37%, and 80.82%,
respectively (Fig.7). After the nitrite treatment, the abundance of functional genes
encoded by *Candidatus* Jettenia increased significantly and became the largest
contributing anammox genus. It indicated that the anammox process was dominated by
*Candidatus* Jettenia bacteria in UASB2, which was consistent with the change in
community structure." (Page 15, Line 370-388)

"Fig. 5. (a) Species richness expressed via Ace and Chao estimator; (b) Species evenness

expressed via the Shannon and Simpson indices; (c) The summary of assembly

processes." (Page 31 line 790-794)

"Fig. 7. The contribution of anammox bacterial communities to key functional genes, the

left side was untreated with nitrite, the right side was treated with nitrite." (Page 33, Line

798-800)

**Several minor remarks:**

1) *“Page 7 line 169: “In phase III (144–184 days), the sludge was returned to each*
*UASB reactor and the temperature was restored to $31.0 \pm 1.5^{\circ}\text{C}$ ” – had the*
*temperature been changed before this time?*

Response:

The sludge was stored at 4°C in a refrigerator before being returned to each
UASB reactor in phase III. This information was shown in the revised manuscript
as follows:

*“After the SAA test, the sludge was starved at 4°C for approximately two weeks.” (Page 7,*
*Line 168)*

2) *Abstract line 18: “reducing potentially pathogenic” here doesn’t fit with the topic*
*of the paper.”*

Response:

We appreciate your observation regarding the term “reducing potentially
pathogenic” in the abstract. As your suggested, we have removed this sentence
from the abstract.

REVIEWER COMMENTS

Reviewer #1 (Remarks to the Author):

In their revised manuscript entitled "Enhancing system antifragility: A novel approach to mitigate High-substrate inhibition in wastewater treatment systems" the authors have made substantial changes and did additional experiments to address the points raised in the first round of review. I do still have questions that I think should be addressed before publication.

- sequenced samples

it is unclear from the revised manuscript sequencing data was added in revision. In line 107-109 of the reviewer response the authors mention frozen samples, but do not state how many, and whether these represent multiple timepoints. This information is also not present in the methods section of the revised manuscript. It would be good to see the results of the additional 16S amplicon sequencing presented as balloon plots or stacked bar charts, perhaps in a supplemental figure, and to have the samples used clarified in the methods.

Related to this point, I can't evaluate whether the community assembly analysis (using icamp) was robust.

It is also unclear which samples were used for shotgun metagenomics, and how deeply these were sequenced. Please clarify this in the methods as well.

- methods citations

most bioinformatics tools are published software, and should be cited. This is true for DIAMOND, KEGG fastp, mothur and qiime (and possibly others in the manuscript). Please check and include citations where appropriate.

- total biomass

This is something that did not occur to me in the first round of review, but as I was considering the community transition between a Kuenenia and Jettenia dominated anammox community, I was wondering whether you quantified total biomass in the reactor, and how much this changed over the course of the experiments? I'm asking because I'm curious how much the relative abundance increase of Jettenia can be attributed to growth off Jettenia, versus die off and washing out of the Kuenenia.

- dissimilatory nitrite reduction to ammonium

you mention in the revised manuscript that the nitrogen removal rate in UASB2 remains high despite the nitrite exceeding the theoretical stoichiometry of the anammox reaction. Elsewhere you state that SJA28 (previously?) had been shown to encode nrfAH genes that can be used to reduce nitrite to ammonium. In principle, this would allow a portion of the nitrite to be converted to ammonium and used in the anammox process if sufficient electron donor is present. Additionally, nitrite (re)formed by nitrate reduction could also be further converted to ammonium this way. Did you check in the metagenomes whether there was a clear difference in reads encoding nrfAH genes? It could furthermore be good to assemble and bin the data to confirm which organisms in the reactors encode which genes, but I understand if this is beyond the scope of this work.

- minor comments:

figure 3 and 7 both have an axis labeled with % as a unit but a value range from 0-1,5 and 0-1 respectively. From context, I think these values should be 0-150 and 0-100.

line 39-40 of the revised manuscript states that nitrification is performed by ammonia-oxidizing bacteria and nitrite-oxidizing bacteria. Ammonia oxidizing archaea should be added for completeness and correctness.

there's a spelling mistake in the link at line 219

line 363: change "the SJA-28 lineage was able to encode" to "In some cases, organisms in the SJA-28 lineage encoded"

line 377-379 misrepresent the role of hydroxylamine in the anammox process. Hydrazine synthase can leak hydroxylamine as a consequence of incomplete hydrazine formation. Hydroxylamine is a strong inhibitor of the anammox process, and in anammox bacteria is detoxified to nitric oxide by hao. please rephrase this sentence

line 399: i'd remove "on day 170" as there is no context provided for the experiment described. Alternatively, explain more clearly

line 412: "transform the anammox genus" is not appropriate, I suggest something like "select for another anammox genus"

Reviewer #2 (Remarks to the Author):

The authors have substantially modified the manuscript, while also including additional data from sequencing and the correction of a number of errors. The manuscript is considerably strengthened as a result and can be considered for publication.

**Detailed response to reviewers' comments**

We would like to thank you for your valuable comments. We have carefully revised
the manuscript considering all these comments. Below are our detailed responses.

**Reviewer #1 (Remarks to the Author):**

*In their revised manuscript entitled "Enhancing system antifragility: A novel*
*approach to mitigate high-substrate inhibition in wastewater treatment systems" the*
*authors have made substantial changes and did additional experiments to address the*
*points raised in the first round of review. I do still have questions that I think should*
*be addressed before publication.*

Response:

We would like to extend our sincere gratitude for your continued evaluation of
our manuscript, and we are grateful for the opportunity to further refine our work.

1. *sequenced samples*

*It is unclear from the revised manuscript sequencing data was added in revision. In*
*line 107-109 of the reviewer response the authors mention frozen samples, but do not*
*state how many, and whether these represent multiple timepoints. This information is*
*also not present in the methods section of the revised manuscript. It would be good to*
*see the results of the additional 16S amplicon sequencing presented as balloon plots*
*or stacked bar charts, perhaps in a supplemental figure, and to have the samples used*
*clarified in the methods. Related to this point, I can't evaluate whether the community*

*assembly analysis (using icamp) was robust.*

*It is also unclear which samples were used for shotgun metagenomics, and how*
*deeply these were sequenced. Please clarify this in the methods as well.*

Response:

We appreciate your attention to the details of our revised manuscript.

The frozen samples referred to in our previous response were collected at the
initial and final moments of UASB1 and UASB2.

There was a total of three samples, and each sample consists of 50 ml. One
samples from the initial moments of two UASB reactors (day 1), were named
"Seed". The samples at the end of the two reactor runs (day 185) were named
"UASB1" and "UASB2", respectively.

Three 16S amplicon sequencing and one metagenomics sequencing were
performed for each sample. Before the first review, all three samples were
subjected to a single amplicon sequencing. In order to understand the genes in the
microbial community, at the same time, to limit the use of tools that try to infer
function from 16S rRNA gene amplicon sequencing, all three samples were
subjected to one metagenomic sequencing. In order to further explore the process
of changing species structure in the reactor over time, all three samples were
again sequenced twice by amplicon sequencing. A total of three sequencing data
were used for community assembly.

The number of valid sequences sequenced for the 16S amplicon of each sample
was 38223, and the size of metagenomic data ranges from 7.3 to 8.8 Gb after

quality control.

We performed balloon plots based on amplicon sequencing results, which were
included in the Supplementary Material as Fig.S2. We have added a description
of the sample time as well as sequencing depth, and updated the methods section
in the revised manuscript:

“2.6. Community structure analysis and visualization

Three sludge samples were collected. One samples from the initial moments of two
UASB reactors (day 1) were named "Seed". Two samples at the end of the two reactor
runs (day 185) were named "UASB1" and "UASB2", respectively. Three replicates of
amplicon sequencing were performed for each sample.” (Page 8, Line 187-191)

“After quality filtering, a mean length ranging from 422 bp were obtained for each sample.
To avoid unequal sequencing depth the number of valid sequences is 38223. These
sequences were clustered into operational taxonomic units (OTUs) at a 97% similarity
level.” (Page 8, Line 197-200)

“2. 7. Functional gene analysis

In order to understand the genes in the microbial community, all three samples were
subjected to one metagenomic sequencing.” (Page 9, Line 208-209)

“The size of metagenomic data of each sample ranges from 7.3 Gb to 8.8 Gb after quality
control.” (Page 9, Line 218-219)

"Fig. S2. Analysis of bacterial community abundance at the genus level from the 16S

amplicon sequencing. Seed-2, UASB-2, and UASB-3 were from the second sequencing,

and Seed-3, UASB-3, and UASB-3 were from the third sequencing. Each sample was

sequenced by three times, and the first sequencing data was shown in Fig.6."

(Supplementary materials)

2. *methods citations*

*Most bioinformatics tools are published software, and should be cited. This is true for*
*DIAMOND, KEGG fastp, mothur and qiime (and possibly others in the manuscript).*
*Please check and include citations where appropriate.*

Response:

We appreciate your attention to the proper citation of bioinformatics tools used in
our research. We have ensured that these citations are now appropriately included
in the methods section of our revised manuscript

"Based on the OTU clustering analysis, Mothur (v.1.30.2) software was used to analyze
the OTU diversity index (alpha diversity analysis within samples)⁵³. Beta diversity was
calculated using QIIME, and hierarchical clustering was performed based on the beta
diversity distance matrix⁵⁴." (Page 8, Line 200-203)

"The functional annotation was conducted using DIAMOND (version 0.8.35) against the
Kyoto Encyclopedia of Genes and Genomes database (<https://www.genome.jp/kegg/>).⁵⁶
⁵⁷" (Page 9, Line 219-222)

53. Schloss PD, et al. Introducing mothur: open-source, platform-independent,
community-supported software for describing and comparing microbial communities.
Appl Environ Microbiol 75, 7537-7541 (2009).

54. Kuczynski J, Stombaugh J, Walters WA, González A, Caporaso JG, Knight R.
Using QIIME to analyze 16S rRNA gene sequences from microbial communities. Curr
Protoc Bioinformatics Chapter 10, 10.17.11-10.17.20 (2011).

56. Buchfink B, Xie C, Huson DH. Fast and sensitive protein alignment using
DIAMOND. Nature Methods 12, 59-60 (2015).

57. Kanehisa M, Goto S. KEGG: kyoto encyclopedia of genes and genomes. Nucleic
Acids Res 28, 27-30 (2000).

3. *total biomass*

*This is something that did not occur to me in the first round of review, but as I was*
*considering the community transition between a Kuenenia and Jettenia dominated*
*anammox community, I was wondering whether you quantified total biomass in the*
*reactor, and how much this changed over the course of the experiments? I'm asking*
*because i'm curious how much the relative abundance increase of Jettenia can be*
*attributed to growth off Jettenia, versus die off and washing out of the kuenenia.*

Response:

Thank you for your insightful question regarding the total biomass in the reactor
and its changes over the course of our experiments. The increase in relative
abundance may be attributed to the fact that the biomass of the anammox bacteria
did not change while the total biomass decreased.

We had quantified the total biomass in the form of mixed liquor suspended solids
(MLSS). The initial MLSS for each reactor was $6.51 \text{ g}\cdot\text{L}^{-1}$ on day 1. At the end of
the experiments (day 185), the MLSS for UASB1 and UASB2 were $7.72 \text{ g}\cdot\text{L}^{-1}$
and $6.93 \text{ g}\cdot\text{L}^{-1}$, respectively. This indicates that the total biomass was increased in
both reactors.

In order to better understand the change in biomass of anammox genera, we
roughly estimated the biomass of anammox genera in the two reactors, by
multiplying the MLSS with the relative abundance of all anammox genera (see
table below).

Sample	Biomass MLSS (g·L ⁻¹)	Relative abundance			Genera biomass (g·L ⁻¹) = biomass * relative abundance			
		Brocadia	Kuenenia	Jettenia	Brocadia	Kuenenia	Jettenia	Total anammox genera
Seed	6.51	0.38%	9.93%	4.90%	0.025	0.646	0.319	0.990
UASB1	7.72	1.25%	12.62%	8.88%	0.097	0.974	0.686	1.756
UASB2	6.93	0.05%	0.02%	28.26%	0.003	0.001	1.958	1.963

The change in biomass of anammox genera = initial biomass + growth – (death +
washing out).

In UASB1, the biomass of all three anammox genera was increased. In UASB2,
the biomass of *Kuenenia* decreased from 0.646 g·L⁻¹ to 0.001 g·L⁻¹ on day 185.

This means that the decrease of biomass caused by the death and washing out was
indeed exist, and it was higher than the growth biomass. The biomass of *Jettenia*
biomass increased from 0.319 g·L⁻¹ to 1.958 g·L⁻¹, indicating that the increase
through growth was much higher than the decrease in biomass caused by the
death and washing out.

We acknowledge that this calculation was not entirely accurate due to the
limitations of MLSS as a measure of total bacterial weight. However, it does
provide a general understanding of the changes in the biomass of anammox
bacteria in the reactors. We hope that this additional information addresses your
concerns.

4. *dissimilatory nitrite reduction to ammonium*

*You mention in the revised manuscript that the nitrogen removal rate in UASB2*
*remains high despite the nitrite exceeding the theoretical stoichiometry of the*
*anammox reaction. Elsewhere you state that SJA28 (previously?) had been shown to*
*encode nrfAH genes that can be used to reduce nitrite to ammonium. In principle, this*
*would allow a portion of the nitrite to be converted to ammonium and used in the*
*anammox process if sufficient electron donor is present. Additionally, nitrite*
*(re)formed by nitrate reduction could also be further converted to ammonium this*
*way.*

*Did you check in the metagenomes whether there was a clear difference in reads*
*encoding nrfAH genes? It could furthermore be good to assemble and bin the data to*
*confirm which organisms in the reactors encode which genes, but I understand if this*
*is beyond the scope of this work.*

Response:

Thank you for your insightful comments regarding dissimilatory nitrite reduction
to ammonium (DNRA), we have conducted a targeted analysis of our
metagenomic data to identify the presence and abundance of *nrfAH* genes.

Our analysis of metagenomic revealed that the *nrfAH* genes were detected in
both reactors, with a higher number and abundance in UASB2 (*nrfA* was 146.12
reads, and *nrfH* was 190.59 reads) compared to UASB1 (*nrfA* was 74.11 reads,
and *nrfH* was 100.78 reads) (Fig. S4). This suggested that the DNRA pathway
may be more active in UASB2, potentially supporting the high nitrogen removal

rates observed in this reactor despite the nitrite exceeding the theoretical
stoichiometry of the anammox reaction.

Furthermore, in our metagenomic data analysis, we traced the contribution of the
specific organisms to *nrfAH* genes. We found that in UASB2, *Jettenia*
contributed the most to the encoding of these genes, with a contribution of 26.45%
to *nrfA* and 30.04% to *nrfH* (Figure S5). In UASB1, *nrfA* was mainly encoded by
*Brocadia* (24.32% contribution) and *nrfH* was mainly encoded by
*unclassified_c__Planctomycetia* (21.49% contribution). And *norank_c__SJA-28*
was poorly distributed in both UASB reactors.

Based on this information, we added a figure for functional gene abundance and
Fig. S5 for *nrfAH* gene contribution to the Supplementary Material:

"Fig. S4. Number of genes reads associated with the anammox process from
metagenomic sequencing." (Supplementary materials)

"Fig. S5. The contribution of bacterial community to *nrfAH* gene, the left side was

untreated with nitrite, the right side was treated with nitrite." (Supplementary materials)

**minor remarks:**

1) *figure 3 and 7 both have an axis labeled with % as a unit but a value range from*
*0-1,5 and 0-1 respectively. From context, I think these values should be 0-150*
*and 0-100.*

Response:

Thank you for bringing this oversight to our attention. We have revised the axis
labels and values not only for Figures 3 and 7, but also for Figures 4 and S1. The
revised figures were as follows:

"Fig. 3. The nitrogen removal performance of effluent nitrite concentrations, NRR and
TN removal efficiency under nitrite shock. (a) was UASB1 reactor and (b) was UASB2
reactor." (Page 29, Line 793-796)

"Fig. 4. Reactor performances of the UASB2 reactor: (a) influent TN concentrations and

NLR; (b) effluent TN concentrations, nitrite exposure concentrations and NRR; (c)

effluent nitrite concentrations, FNA and TN removal efficiency." (Page 30, Line 797-800)

"Fig. 7. The contribution of anammox bacterial community to key functional genes, the

left side was untreated with nitrite, the right side was treated with nitrite." (Page 33, Line

808-810)

"Fig. S1. Reactor performances of the UASB1 reactor: (a) influent TN concentrations and

NLR; (b) effluent TN concentrations and NRR; (c) effluent nitrite concentrations, FNA

and TN removal efficiency." (Supplementary materials)

2) *line 39-40 of the revised manuscript states that nitrification is performed by*

ammonia-oxidizing bacteria and nitrite-oxidizing bacteria. Ammonia oxidizing

archaea should be added for completeness and correctness.

Response:

Thank you for your careful review and we revised the manuscript. The revised

manuscript as follows:

"The process of nitrification involves the oxidation of ammonia to nitrate by
ammonia-oxidizing bacteria, ammonia oxidizing archaea and nitrite-oxidizing bacteria"
(Page 2, Line 38-40)

3) *there's a spelling mistake in the link at line 219.*

Response:

We have corrected the spelling error in the link. The revised sentence is as
follows:

"The functional annotation was conducted using DIAMOND (version 0.8.35) against the
Kyoto Encyclopedia of Genes and Genomes database (<https://www.genome.jp/keeg/>)."
(Page 9, Line 219-222)

4) *line 363: change "the SJA-28 lineage was able to encode" to "In some cases,*
*organisms in the SJA-28 lineage encoded".*

Response:

We have made the requested change to line 363 as follows:

"In some cases, organisms in the SJA-28 lineage encoded an ammonia-forming
cytochrome c nitrite reductase of the nrfAH-type involved in nitrite reduction." (Page 15,
Line 359-361)

5) *line 377-379 misrepresent the role of hydroxylamine in the anammox process.*
*Hydrazine synthase can leak hydroxylamine as a consequence of incomplete*
*hydrazine formation. Hydroxylamine is a strong inhibitor of the anammox process,*
*and in anammox bacteria is detoxified to nitric oxide by hao. please rephrase this*
*sentence.*

**Response:**

Thank you for pointing out the misrepresentation. We have revised the
manuscript as follows:

"HAO was believed to reconvert any hydroxylamine (NH₂OH) that escaped from the HZS
complex back into nitric oxide, thereby detoxifying this inhibitor NH₂OH and ensuring
the normal progression of the anammox process.^{75, 76}" (Page 15, Line 374-377)

75. Akram M, Dietl A, Müller M, Barends TRM. Purification of the key enzyme
complexes of the anammox pathway from DEMON sludge. *Biopolymers* 112, (2021).

76. Maalcke WJ, et al. Structural Basis of Biological NO Generation by Octaheme
Oxidoreductases. *Journal of Biological Chemistry* 289, 1228-1242 (2014).

6) *line 399: i'd remove "on day 170" as there is no context provided for the*
*experiment described. Alternatively, explain more clearly*

**Response:**

We have removed "on day 170" at line 399 of our manuscript.

7) *line 412: "transform the anammox genus" is not appropriate, I suggest something*
*like "select for another anammox genus"*

Response:

We have revised the manuscript as follows:

"This operation is not sufficient to select for another anammox genus into one that is more
adapted to high nitrite concentrations" (Page 17, Line 409-410)

REVIEWERS' COMMENTS

Reviewer #1 (Remarks to the Author):

With their latest revisions, the authors have sufficiently addressed my earlier questions and can be considered for publication

One tiny thing, the link on line 221 is still incorrect. It should read:
<https://www.genome.jp/kegg/>

Detailed response to reviewers' comments

We would like to thank you for your valuable comments. We have carefully revised the manuscript considering all these comments. Below are our detailed responses.

Reviewer #1 (Remarks to the Author):

With their latest revisions, the authors have sufficiently addressed my earlier questions and can be considered for publication

One tiny thing, the link on line 221 is still incorrect. It should read:

<https://www.genome.jp/kegg/>

Response:

Thank you for your positive feedback regarding our revised manuscript and for pointing out the incorrect link on line 221. We have updated the link on line 221 to the correct URL:

“The functional annotation was conducted using DIAMOND (version 0.8.35) against the Kyoto Encyclopedia of Genes and Genomes database (<https://www.genome.jp/kegg/>).”

(Page 19, Line 460-462)